# Tackling the Data Heterogeneity in Asynchronous Federated Learning with Cached Update Calibration

**Yujia Wang**[1], **Yuanpu Cao**[1], **Jingcheng Wu**[2], **Ruoyu Chen**[2], **Jinghui Chen**[1]

[1]The Pennsylvania State University [2]Carnegie Mellon University

`{yjw5427, ymc5533}@psu.edu`, `{jingchew, ruoyuche}@andrew.cmu.edu`,
`jzc5917@psu.edu`

## Abstract

Asynchronous federated learning, which enables local clients to send their model update asynchronously to the server without waiting for others, has recently emerged for its improved efficiency and scalability over traditional synchronized federated learning. In this paper, we study how the asynchronous delay affects the convergence of asynchronous federated learning under non-i.i.d. distributed data across clients. Through the theoretical convergence analysis of one representative asynchronous federated learning algorithm under standard nonconvex stochastic settings, we show that the asynchronous delay can largely slow down the convergence, especially with high data heterogeneity. To further improve the convergence of asynchronous federated learning under heterogeneous data distributions, we propose a novel asynchronous federated learning method with a cached update calibration. Specifically, we let the server cache the latest update for each client and reuse these variables for calibrating the global update at each round. We theoretically prove the convergence acceleration for our proposed method under nonconvex stochastic settings. Extensive experiments on several vision and language tasks demonstrate our superior performances compared to other asynchronous federated learning baselines.

## 1 Introduction

Federated learning (FL) (Konečný et al., 2016) has become an increasingly popular large-scale machine learning paradigm where machine learning models are trained on multiple edge clients guided by a central server. FedAvg (McMahan et al., 2017) (McMahan et al., 2017), also known as Local SGD (Stich, 2018), is one of the most popular federated optimization methods, where each client locally performs multiple steps of SGD updates followed by the synchronous server aggregation of the local models. However, the traditional synchronous aggregation scheme may cause efficiency and scalability issues as the server need to wait for all participating clients to complete the task before conducting the global update step. This promotes the development of asynchronous federated learning methods such as FedAsync (Xie et al., 2019), FedBuff (Nguyen et al., 2022), which adopt flexible aggregation schemes and allow clients to asynchronously send back their model update and thus improve the overall training efficiency and scalability.

Such an asynchronous aggregation scheme does not come with no costs: the asynchronous delay, which describes the fact that the delayed local model update could be computed based on a past global model rather than the current global model, slows down the convergence of asynchronous federated learning. Moreover, the negative impact of the asynchronous delay on the convergence gets even worse when the training data are non-i.i.d. distributed across clients. This is intuitive since empirical observation suggests that the global model changes more significantly in adjacent rounds when the data heterogeneity is high. Consequently, the asynchronous delay would cause the delayed local model update to be more outdated and inconsistent with the current global model, hence worsening the overall model convergence. Furthermore, each global update in asynchronous federated learning

is, by nature, contributed from only a fraction of clients (similar to the partial participation scenarios in synchronous FL). This intensifies the global variance arising from data heterogeneity and leads to a further slowdown in convergence. Therefore, it is crucial to tackle the data heterogeneity issue to improve the overall convergence of asynchronous federated learning.

In this work, we rigorously study how the asynchronous delay affects the convergence of asynchronous federated learning under non-i.i.d. distributed data across clients. Through a theoretical convergence analysis of FedBuff (Nguyen et al., 2022)[1], one representative asynchronous federated learning algorithm, we show that the asynchronous delay can largely slow down the convergence, especially with high data heterogeneity. To further improve the convergence of asynchronous federated learning under heterogeneous data distributions, we propose a novel asynchronous federated learning method, Cache-Aided Asynchronous Federated Learning (CA$^2$FL). CA$^2$FL allows the server to cache the latest update from each client and reuses this cached update for calibrating the global update, which does not incur extra communication/computation overhead on clients, or raise any additional privacy concerns (same as the traditional synchronous and asynchronous federated learning methods). We summarize our contribution in this paper as follows:

- We present a convergence analysis of FedBuff (Nguyen et al., 2022), one representative asynchronous federated learning algorithm, under non-i.i.d. distributed data across clients (with fewer assumptions and slightly tighter bound on the asynchronous delay term). We demonstrate that the asynchronous delay can theoretically slow down the convergence and such an impact could be further amplified by the highly non-i.i.d. distributed data.

- To tackle the convergence degradation in asynchronous federated learning caused by the joint effect of data heterogeneity and asynchronous delay, we propose a novel asynchronous federated aggregation method with cached update calibrations (CA$^2$FL) in which the server maintains cache updates for each client and reuse the cached update for global aggregation calibration. We theoretically show that with the help of cached updates, our proposed method can significantly improve the convergence rate under nonconvex stochastic settings.

- Extensive experiments on several vision and language tasks demonstrate our superior performances compared to other asynchronous federated learning baselines and back up our theory.

## 2  RELATED WORK

**Synchronous FL and Heterogeneity Issues.**  Federated learning (Konečný et al., 2016) plays a critical role in jointly training models at edge devices without sharing local data. Since FedAvg (McMahan et al., 2017), many federated learning variants are proposed (Li et al., 2019b; Stich, 2018; Yang et al., 2021) for various training scenarios. Reddi et al. (2021); Tong et al. (2020); Wang et al. (2022) propose adaptive federated optimizers for dealing with heavy-tail stochastic gradient noise distributions. Gu et al. (2021); Yan et al. (2020) focus on improving the overall FL performance by leveraging the latest historical gradients. Recently, many works also focused on addressing the data heterogeneity issue through several aspects. FedProx (Li et al., 2020) adds a proximate term to align the local model with the global one. FedDyn (Acar et al., 2021) involves a dynamic regularization term for local and global model consistency. FedNova (Wang et al., 2020b) proposes a normalized averaging mechanism that reduces objective inconsistency with heterogeneous data. Moreover, several works studied how to eliminate the client drift caused by data heterogeneity from the aspect of variance reduction including Karimireddy et al. (2020b;a); Khanduri et al. (2021); Cutkosky & Orabona (2019); Jhunjhunwala et al. (2022). They usually introduce additional control variables to track and correct the local model shift during local training, at the cost of extra communications for synchronizing these control variables. Besides, FedDC (Gao et al., 2022) involves both dynamic regularization terms and local drift variables for model correction.

**Asynchronous SGD and Asynchronous FL.**  Asynchronous optimization methods such as asynchronous SGD and its variants have been discussed for many years. Hogwild! SGD (Niu et al., 2011) studies a coordinate-wise asynchronous method without any locking, and (Nguyen et al., 2018)

---

[1]Here we focus on the FedBuff algorithm without differential privacy for the entire paper.

provided a tight convergence analysis for SGD and Hogwild! algorithm. Some other works focus on the theoretical analysis for the asynchronous SGD such as (Mania et al., 2017; Stich et al., 2021). (Leblond et al., 2018) studies the asynchronous SAGA method and demonstrates its theoretical convergence. (Glasgow & Wootters, 2022) explored asynchronous SAGA methods for the distributed-data setting and provided a theoretical analysis. In the context of federated learning, FedAsync (Xie et al., 2019) is proposed for clients to update asynchronously to the server. FedBuff (Nguyen et al., 2022) proposed a buffered asynchronous aggregation strategy. Later Toghani & Uribe (2022) studies the convergence analysis of FedBuff with fewer assumptions. Anarchic Federated Averaging (Yang et al., 2022) focuses on letting the clients decide when and whether to participate in global training. Stripelis et al. (2022) proposed a semi-synchronous federated learning method for energy-efficient training and accelerating convergence in cross-silo settings. SWIFT (Bornstein et al., 2023) is an interesting wait-free decentralized FL paradigm that shares a similar idea to asynchronous FL, and SWIFT also involves caching models by storing the neighboring local models. Moreover, there are several works studying the theoretical convergence analysis in asynchronous federated learning with arbitrary delay (Avdiukhin & Kasiviswanathan, 2021; Mishchenko et al., 2022) or the complete theoretical analysis under various assumptions (Koloskova et al., 2022).

## 3    PRELIMINARIES FINDINGS ON ASYNCHRONOUS FEDERATED LEARNING

**Federated Learning.**  In general federated learning framework, we aim to minimize the following objective through $N$ local clients:

$$\min_{\boldsymbol{x} \in \mathbb{R}^d} f(\boldsymbol{x}) := \frac{1}{N} \sum_{i=1}^{N} F_i(\boldsymbol{x}) = \frac{1}{N} \sum_{i=1}^{N} \mathbb{E}_{\xi \sim \mathcal{D}_i}[F_i(\boldsymbol{x}; \xi_i)], \tag{3.1}$$

where $\boldsymbol{x}$ represents the model parameters with $d$ dimensions, $F_i(\boldsymbol{x}) = \mathbb{E}_{\xi \sim \mathcal{D}_i}[F_i(\boldsymbol{x}, \xi_i)]$ represents the local loss function corresponding to client $i$ and let $\mathcal{D}_i$ denotes the local data distribution on client $i$. FedAvg (McMahan et al., 2017) is a popular synchronous optimization algorithm to solve Eq. 3.1, where each participating client performs local SGD updates, and the server performs global averaging steps after receiving all the updates from assigned clients.

**Asynchronous Federated Learning.**   Asynchronous federated learning has been introduced to facilitate efficiency and scalability for clients in solving Eq. 3.1 asynchronously. In asynchronous federated learning, clients are allowed to train and synchronize local models on their own pace. For example, FedBuff (Nguyen et al., 2022) studied an asynchronous federated learning method with a global update buffer and differential privacy mechanism. To give a more concrete idea, we present a general asynchronous federated learning framework, which is essentially FedBuff without the differential privacy part, as shown in Algorithm 1. Specifically, the server initializes by randomly selecting an active client set $\mathcal{M}_1$ with the size of the concurrency[2] $M_c$. Then each assigned client will conduct $K$ steps of local training asynchronously. This means the server does not need to wait until all assigned clients finish their local training to proceed, instead, the server just accumulates the model update in $\boldsymbol{\Delta}_t$ (Line 5 in Algorithm 1) and updates the global model every time it accumulates $M$ updates[3] (Lines 9-11 in Algorithm 1). Meanwhile, once the server receives a client update, it will instantly re-sample another available client to continue the federated learning procedure. In this way, the server always maintains a fixed number of active clients (i.e., the concurrency $M_c$).

**Heterogeneity Across Clients.**  Several works (Karimireddy et al., 2020b;a; Acar et al., 2021; Wang et al., 2020b) have shown that synchronized federated learning methods suffer from convergence and empirical degradation when data is heterogeneously distributed across local clients. This issue of model inconsistency also occurs in asynchronous federated learning and may even become worse with the existing of gradient delay, since the model used for local gradient computation is usually different from the current global model, which makes local updates less representative of the global update direction.  In order to formally illustrate such a relationship, we conduct the following convergence analysis on Algorithm 1 under standard stochastic nonconvex optimization settings. First, we introduce some necessary assumptions.

---

[2]The concurrency implies that the maximum size of the simultaneously active clients is $M_c$.

[3]$M$ denotes the buffer size as in Fedbuff and $M \geq 1$.

---

**Algorithm 1** FedBuff without DP

---

**Input:** local step size $\eta_l$, global stepsize $\eta$, server concurrency $M_c$, buffer size $M$;

1: Initialize $\boldsymbol{\Delta}_1 = \mathbf{0}, m = 0$ and sample a set of $M_c$ active clients to run local SGD updates.
2: **repeat**
3:     **if** receive client update **then**
4:         Server accumulates update from client $i$: $\boldsymbol{\Delta}_t \leftarrow \boldsymbol{\Delta}_t + \boldsymbol{\Delta}_t^i$ and set $m \leftarrow m + 1$
5:         Samples another client $j$ from available clients
6:         Broadcast the current model $\boldsymbol{x}_t$ to client $j$, and run local SGD updates on client $j$
7:     **end if**
8:     **if** $m = M$ **then**
9:         Update global model $\boldsymbol{x}_{t+1} = \boldsymbol{x}_t + \eta \frac{\boldsymbol{\Delta}_t}{M}$
10:        Set $m \leftarrow 0, \boldsymbol{\Delta}_{t+1} \leftarrow \mathbf{0}, t \leftarrow t + 1$
11:     **end if**
12: **until** Convergence

---

**Assumption 3.1** (Smoothness). Each loss function on the $i$-th worker $F_i(\boldsymbol{x})$ is $L$-smooth, i.e., $\forall \boldsymbol{x}, \boldsymbol{y} \in \mathbb{R}^d$,

$$\left| F_i(\boldsymbol{x}) - F_i(\boldsymbol{y}) - \langle \nabla F_i(\boldsymbol{y}), \boldsymbol{x} - \boldsymbol{y} \rangle \right| \leq \frac{L}{2} \|\boldsymbol{x} - \boldsymbol{y}\|^2.$$

This also implies the $L$-gradient Lipschitz condition, i.e., $\|\nabla F_i(\boldsymbol{x}) - \nabla F_i(\boldsymbol{y})\| \leq L\|\boldsymbol{x} - \boldsymbol{y}\|$. Assumption 3.1 is a standard assumption in nonconvex optimization problems, which has been also adopted in (Kingma & Ba, 2015; Reddi et al., 2018; Li et al., 2019a; Yang et al., 2021).

**Assumption 3.2** (Bounded Variance). Each stochastic gradient on the $i$-th worker has a bounded local variance, i.e., for all $\boldsymbol{x}, i \in [N]$, we have $\mathbb{E}[\|\nabla f_i(\boldsymbol{x}, \xi) - \nabla F_i(\boldsymbol{x})\|^2] \leq \sigma^2$, and the loss function on each worker has a global variance bound, $\frac{1}{N} \sum_{i=1}^N \|\nabla F_i(\boldsymbol{x}) - \nabla f(\boldsymbol{x})\|^2 \leq \sigma_g^2$.

Assumption 3.2 is widely used in federated optimization problems (Li et al., 2019a; Reddi et al., 2021; Yang et al., 2021). The bounded local variance represents the randomness of stochastic gradients, and the bounded global variance represents data heterogeneity between clients. Note that $\sigma_g = 0$ corresponds to the *i.i.d* setting, in which datasets from each client have the same distribution.

**Assumption 3.3** (Bounded Gradient Delay). Let $\tau_t^i$ represent the delay for global round $t$ and client $i$ which is applied in Algorithm 1 and 2. $\tau_t^i$ implies the difference between the current global round $t$ and the global round at which client $i$ started to compute the gradient. We assume that the maximum gradient delay is bounded, i.e., $\tau_{\max} = \max_{t \in [T], i \in [N]}\{\tau_t^i\} < \infty$.

Assumption 3.3 is a common assumption in convergence analysis for asynchronous federated learning method (Koloskova et al., 2022; Yang et al., 2020). Note that Assumption 3.3 naturally means that the average delay $\tau_{\text{avg}} = \frac{1}{NT} \sum_{t=1}^T \sum_{i=1}^N \tau_t^i < \infty$ is bounded.

**Theorem 3.4.** Under Assumptions 3.1-3.3, denote $f_* = \arg\min_{\boldsymbol{x}} f(\boldsymbol{x})$ and $f_1 = f(\boldsymbol{x}_1)$, let $T$ be the total global rounds and $K$ be the number of local SGD training steps. If the local learning rate $\eta = \Theta(\sqrt{KM})$ and $\eta_l = \Theta(1/\sqrt{T}K)$ then the global rounds of Algorithm 1 satisfy

$$\frac{1}{T} \sum_{t=1}^T \mathbb{E}[\|\nabla f(\boldsymbol{x}_t)\|^2] = \mathcal{O}\left( \frac{[(f_1 - f_*) + \sigma^2]}{\sqrt{TKM}} \right) + \mathcal{O}\left( \frac{\sigma^2 + K\sigma_g^2}{TK} \right)$$

$$+ \mathcal{O}\left( \frac{\sqrt{K}}{\sqrt{TM}} \sigma_g^2 \right) + \mathcal{O}\left( \frac{K\tau_{\max}\tau_{\text{avg}}\sigma_g^2 + \tau_{\max}\sigma^2}{T} \right). \tag{3.2}$$

**Remark 3.5.** Theorem 3.4 presents the convergence analysis for Algorithm 1 w.r.t. global communication round $T$, local steps $K$ and the update accumulation amount $M$. From Eq. equation 3.2, it can be seen that the maximum delay $\tau_{\max}$ and the average delay $\tau_{\text{avg}}$ term indeed affects the overall convergence of the asynchronous federated learning algorithm. Particularly, the last term involves joint effect term $\mathcal{O}(K\tau_{\max}\tau_{\text{avg}}\sigma_g^2/T)$ where the global variance $\sigma_g^2$ and the delay terms $\tau_{\max}$ and $\tau_{\text{avg}}$ are multiplied together. This implies that the convergence degradation brought by the

asynchronous delay is amplified by the high data heterogeneity (large $\sigma_g$). If data are i.i.d. distributed across clients, i.e., $\sigma_g = 0$, then $\mathcal{O}(K\tau_{\max}\tau_{\text{avg}}\sigma_g^2/T)$ term vanishes to 0. On the other hand, if data are non-i.i.d. distributed, i.e., $\sigma_g \neq 0$, the term $\mathcal{O}(K\tau_{\max}\tau_{\text{avg}}\sigma_g^2/T)$ will largely slow down the overall convergence (in fact, when $T \leq KM$, this term would become the dominant term in the convergence rate). This verifies our intuition that the data heterogeneity can worsen the impact of asynchronous delay and jointly deteriorate the convergence, which motivates us to develop a novel method for reducing such joint effects and improving the convergence for asynchronous federated learning. Compared to the original analysis in FedBuff, our analysis requires fewer assumptions and enjoys a slightly tighter bound on the asynchronous delay term[4].

## 4 PROPOSED METHOD: CACHE-AIDED ASYNCHRONOUS FL

To address the challenges of data heterogeneity and gradient delay across clients and achieve better convergence in asynchronous federated learning, we propose a novel Cache-Aided Asynchronous FL (CA$^2$FL) method. The proposed CA$^2$FL enables the server to maintain and reuse the cached updates for global update calibration. Algorithm 2 summarizes our proposed CA$^2$FL. In general, the CA$^2$FL largely follows the FedBuff framework in Algorithm 1, while the main difference between our proposed CA$^2$FL and Algorithm 1 lies primarily in the global update steps. Specifically, we introduce a *cached variable updating* shown in Line 5 and 13, and we incorporate a *global calibration* process in Line 4 and 11.

---

**Algorithm 2** Cached-Aided Asynchronous FL

**Input:** local step size $\eta_l$, global stepsize $\eta$, server concurrency $M_c$, buffer size $M$;

1: Initialize $\boldsymbol{\Delta}_1 = \mathbf{0}, \boldsymbol{h}_1^i = \mathbf{0}$ for $i \in [N]$, $\boldsymbol{h}_1 = \mathbf{0}$, $m = 0$ and sample a set of $M_c$ active clients to run local SGD updates.
2: **repeat**
3:     **if** receive client update **then**
4:         Server accumulates calibrated update from client $i$: $\boldsymbol{\Delta}_t \leftarrow \boldsymbol{\Delta}_t + (\boldsymbol{\Delta}_t^i - \boldsymbol{h}_t^i)$
5:         Server update clients' cached variables: $\boldsymbol{h}_{t+1}^i = \boldsymbol{\Delta}_t^i$
6:         Set $m \leftarrow m + 1$, $\mathcal{S}_t \leftarrow \mathcal{S}_t \cup \{i\}$
7:         Samples another client $j$ from available clients
8:         Broadcast the current model $\boldsymbol{x}_t$ to client $j$, and run local SGD updates on client $j$
9:     **end if**
10:    **if** $m = M$ **then**
11:       $\boldsymbol{v}_t = \boldsymbol{h}_t + \frac{1}{|\mathcal{S}_t|}\boldsymbol{\Delta}_t$
12:       Update global model $\boldsymbol{x}_{t+1} = \boldsymbol{x}_t + \eta\boldsymbol{v}_t$
13:       Server maintains the cached variable $\boldsymbol{h}_{t+1}^i = \boldsymbol{h}_t^i$ for $i \notin \mathcal{S}_t$
14:       Server initialize $\boldsymbol{h}_{t+1} = \frac{1}{N}\sum_{i=1}^N \boldsymbol{h}_{t+1}^i$
15:       Set $m \leftarrow 0$, $\boldsymbol{\Delta}_{t+1} \leftarrow \mathbf{0}$, $\mathcal{S}_{t+1} \leftarrow \emptyset$, $t \leftarrow t + 1$,
16:    **end if**
17: **until** Convergence

---

**Cached variable update.** In CA$^2$FL, the server maintains the latest cached update for each client, and reuses this cached update as an approximation of each client's contribution to the current round's update. Denote $\boldsymbol{h}_t^i$ as the latest cached variable for client $i$ and $\boldsymbol{h}_t$ as the global cached variable which is the average of $\boldsymbol{h}_t^i$ among all clients, i.e., $\boldsymbol{h}_t = \frac{1}{N}\sum_{i=1}^N \boldsymbol{h}_t^i$. Once the server received $\boldsymbol{\Delta}_t^i$ from client $i$, then the server updates the cached variable for it, i.e., $\boldsymbol{h}_{t+1}^i = \boldsymbol{\Delta}_t^i$ (Line 6). For clients which don't contribute to round $t$, the server keeps the state variable unchanged as $\boldsymbol{h}_{t+1}^i = \boldsymbol{h}_t^i$ (Line 14). This update rule for cached variable enforces the server maintains the latest model update difference for each client for global update calibration.

---

[4]Due to space limitations, we leave further discussions about Theorem 3.4 and the comparison with FedBuff analysis in (Toghani & Uribe, 2022) in Appendix B.

**Global calibration.** Once client $i$ finish the local training and send $\boldsymbol{\Delta}_t^i$ to the server, the server accumulates $\boldsymbol{\Delta}_t^i - \boldsymbol{h}_t^i$ to $\boldsymbol{\Delta}_t$. Let $\mathcal{S}_t$ represent a set of clients in which the server received their update at round $t$. After the server receives $M$ updates, we calculate the server updates $\boldsymbol{v}_t$ as the summation of the global cached variable $\boldsymbol{h}_t$ with the average global update $\frac{\boldsymbol{\Delta}_t}{|\mathcal{S}_t|}$ (Line 11). The global model $\boldsymbol{x}_{t+1}$ is then updated by this calibrated variable $\boldsymbol{v}_t$. Note that $\boldsymbol{v}_t$ is actually a linear combination in terms of the latest received model update difference $\boldsymbol{\Delta}_t^i$ and cached variable $\boldsymbol{h}_t$, i.e.,

$$\boldsymbol{v}_t = \boldsymbol{h}_t + \frac{1}{|\mathcal{S}_t|} \sum_{i \in \mathcal{S}_t} (\boldsymbol{\Delta}_t^i - \boldsymbol{h}_t^i). \tag{4.1}$$

**Discussion.** The design (Eq. 4.1) for the calibration and cached variables felt somewhat similar to SAGA (Defazio et al., 2014), a well-recognized stochastic variance-reduction method that stores previously computed gradients and leverages them for reducing the gradient variance. Eq. 4.1 looks like a special form of SAGA by treating model update difference $\boldsymbol{\Delta}_t^i$ as gradients and applied globally over different clients. However, it is important to note that our method does not adhere to the properties of unbiased incremental gradients that SAGA mainly relies on for its variance reduction purposes, which makes our theoretical analysis non-trial and different from that of SAGA. Therefore, CA$^2$FL should not be considered as a direct application of SAGA to asynchronous federated learning. Note that CA$^2$FL does not require extra communication and computation overhead on clients, and it is compatible with privacy persevering approaches such as differential privacy and secure aggregation.

## 5 CONVERGENCE ANALYSIS

We first introduce the additional assumption needed for the convergence analysis of our proposed CA$^2$FL algorithm.

**Assumption 5.1** (Bounded State Delay). Let $\zeta_t^j$ represent the delay of the state variable for global round $t$ and client $j \notin \mathcal{S}_t$ in Algorithm 2. $\zeta_t^j$ is state in the context of client $j$ which does not update the model difference in round $t$ and then maintains the state variable $\boldsymbol{h}_t^j$ as the last step. $\zeta_t^j$ implies the difference between the current global round $t$ and the global round at which this client $j$ started to compute the last gradient. We assume that the maximum gradient delay is also bounded, i.e., $\zeta_{\max} = \max_{t \in [T], j \in [N]} \{\zeta_t^j\} < \infty$.

Assumption 5.1 is also commonly used in convergence analysis for memory-aided federated learning method (Gu et al., 2021; Yang et al., 2022). In a nutshell, the state delay describes how many global rounds has it been since the last local training for a client. In the following, we will show the convergence results for our proposed CA$^2$FL.

**Theorem 5.2.** Under Assumptions 3.1-3.3 and Assumption 5.1, if the local learning rate $\eta = \Theta(\sqrt{KM})$ and $\eta_l = \Theta(1/\sqrt{TK})$ then the global rounds of Algorithm 2 satisfy

$$\frac{1}{T} \sum_{t=1}^{T} \mathbb{E}[\|\nabla f(\boldsymbol{x}_t)\|^2] = \mathcal{O}\left(\frac{f_1 - f_*}{\sqrt{TKM}}\right) + \mathcal{O}\left(\frac{\sigma^2}{\sqrt{TKM}}\right) + \mathcal{O}\left(\frac{\sigma^2 + K\sigma_g^2}{TK}\right) + \mathcal{O}\left(\frac{(\tau_{\max} + \zeta_{\max})\sigma^2}{T}\right),$$
$$\tag{5.1}$$

where $f_* = \arg\min_{\boldsymbol{x}} f(\boldsymbol{x})$.

**Remark 5.3.** Theorem 5.2 suggests that with a sufficient amount of global rounds $T$, i.e., $T \geq KM$, our proposed CA$^2$FL method achieves a desired convergence rate of $\mathcal{O}(\frac{1}{\sqrt{TKM}})$ w.r.t. global round $T$, local steps $K$ and the update accumulation amount $M$, which matches the convergence rate in traditional synchronous federated learning baselines (Yang et al., 2021; Reddi et al., 2021; Jhunjhunwala et al., 2022).

**Remark 5.4.** Compared with Eq. 3.2, the joint effect term $\mathcal{O}(K\tau_{\max}\tau_{\text{avg}}\sigma_g^2/T)$ no longer exists, while in Eq. 5.1, the asynchronous delay $\tau_{\max}$ only relates to the stochastic noise $\sigma$. This suggests that our proposed CA$^2$FL can benefit from the design of reusing the cached update for global update calibration, which tackles the data heterogeneity issue across clients and reduces the joint impact caused by the asynchronous delay and data heterogeneity. Note that our design also contributes to the general data heterogeneity issue in that the $\mathcal{O}(\frac{\sqrt{K}}{\sqrt{TM}}\sigma_g^2)$ term in Eq. 3.2 also gets smaller. Together, those two improvements finally lead to a better convergence rate for our proposed CA$^2$FL algorithm.

## 6  EXPERIMENTAL RESULTS

**Datasets, models, and methods.**  We present the experimental results on both vision and language tasks to verify the effectiveness of the proposed method. For the vision tasks, we train the CIFAR-10 dataset with CNN (Wang & Ji, 2022) and ResNet-18 (He et al., 2016) models, and we also train CIFAR-100 (Krizhevsky et al., 2009) datasets with ResNet-18 model, and we provide various data sampling levels and client concurrency settings. For the language tasks, we conduct experiments on fine-tuning a pretrained Bert-base model (Devlin et al., 2018) on several datasets in GLUE benchmark (Wang et al., 2018). We evaluate experiments on non-i.i.d. data distributions by a Dirichlet distribution partitioned strategy similar to (Wang et al., 2020a;b) with several parameters for both vision and language tasks. We adopt the same CNN network as in and ResNet-18 network (He et al., 2016). We compare our proposed CA$^2$FL with the asynchronous FL baseline, FedBuff (without differential privacy) (Nguyen et al., 2022) and FedAsync Xie et al. (2019) (constant), and with the synchronized FL method, FedAvg McMahan et al. (2017). Due to the space limit, we leave additional experiments on more datasets and models together with the experiment details in Appendix A [5].

**Implementation overview of vision tasks.**  For experiments on CIFAR-10 and CIFAR-100, the number of local training iterations $K$ on each client is set to two local epochs (the amount of iteration depends on the amount of data for each client, and the batch size is set to 50 for all experiments by default). For local update, we use the SGD optimizer with a learning rate gridding from {0.001, 0.01, 0.1, 1} with momentum 0.9 and weight decay of `1e-4`, and the global learning rate is gridding from {0.1, 1.0, 2.0} for all methods. We set a total of 100 clients in the network and the concurrency $M_c = 20$ if there is no further instructions, and we set the update accumulation amount $M = 10$ by default.

**Implementation overview of language tasks.**  For experiments on fine-tuning Bert-base model on the MRPC, SST-2, RTE and CoLA datasets from the GLUE benchmark, the number of local iterations $K$ on each client is one local epoch (the amount of iteration depends on the amount of data for each client, and the batch size is set to 32 for all experiments by default). We employ Dir (0.6) for non-i.i.d. data partitioned among clients. We adopt the low-rank adaptation (LoRA) (Hu et al., 2021) as the parameter-efficient fine-tuning method. Specifically, for a pre-trained weight matrix $W_0 \in \mathbb{R}^{d \times k}$, LoRA freezes $W_0$ but tuned the $\Delta W$ by representing with a low-rank decomposition with rank $r \ll \min(d, k)$, $W_0 + \Delta W = W_0 + BA$, where $B \in \mathbb{R}^{d \times r}$ and $A \in \mathbb{R}^{r \times k}$ are two trainable parameters. For all experiments, we choose $r = 1$ and $\alpha_{\text{LoRA}} = 1$. For local update, we use the widely-used AdamW optimizer with a learning rate gridding from {`5e-5, 1e-4, 5e-4, 1e-3 5e-3`} with weight decay of `1e-4`, and the global learning rate is gridding from {0.1, 1} for all methods. We set a total of 10 clients in the network and the concurrency $M_c = 5$ if there are no further instructions, and we set the update accumulation amount $M = 3$ by default.

### 6.1  MAIN RESULTS

Table 1 shows the overall performance of training CIFAR-10 with a CNN model and the ResNet-18 model. We observe that the proposed CA$^2$FL shows improvement upon the FedBuff and FedAsync. Particularly, when training with the lightweight CNN model (with about 2.2M trainable parameters), the training loss of FedAsync is severely fluctuating and cannot converge when $\alpha = 0.1$, while our proposed CA$^2$FL are more robust to the highly heterogeneous settings and achieve better result than FedBuff.

Table 2 presents the overall test accuracy of experiments on CIFAR-100 with two data heterogeneity levels. For $\alpha = 0.1$, our proposed CA$^2$FL achieves higher test accuracy compared to FedBuff but has lower accuracy than FedAsync. Specifically, when the data is highly heterogeneously distributed, e.g., $\alpha = 0.01$, our CA$^2$FL method significantly outperforms than FedBuff and FedAsync.

For fine-tuning the Bert-base model on the GLUE benchmark, Table 3 presents the evaluation results for four datasets with several tasks. Note that the MRPC, SST-2, and RTE datasets are evaluated

---

[5]We also provide experimental results on fine-tuning TinyImageNet with two ResNet models, and parameter-efficient fine-tuning GPT-2 small model on E2E NLG Challenge.

Table 1: The test accuracy of different models on the CIFAR-10 dataset with different models and data heterogeneity degrees. We report the mean accuracy and the standard derivation for the last 5 rounds.

| Method | Dir(0.3) | | Dir(0.1) | |
|---|---|---|---|---|
| | CNN Acc. & std | ResNet-18 Acc. & std | CNN Acc. & std | ResNet-18 Acc. & std |
| FedAsync | $62.29 \pm 0.16$ | $79.8 \pm 2.28$ | - | $40.58 \pm 2.92$ |
| FedBuff | $60.74 \pm 1.18$ | $78.53 \pm 3.31$ | $53.96 \pm 0.10$ | $63.03 \pm 3.17$ |
| CA$^2$FL | $\mathbf{64.40} \pm 0.32$ | $\mathbf{83.79} \pm 0.34$ | $\mathbf{57.62} \pm 0.42$ | $\mathbf{68.37} \pm 1.97$ |

by the validation accuracy, while the CoLA dataset is evaluated by Matthew's correlation. We observe that FedAsync achieves higher validation accuracy in MRPC, for other tasks and datasets, our proposed CA$^2$FL obtains better evaluation results. Moreover, we plot the training loss w.r.t. the global rounds in Figure 1, and it verifies the theoretical convergence improvements of our proposed CA$^2$FL.

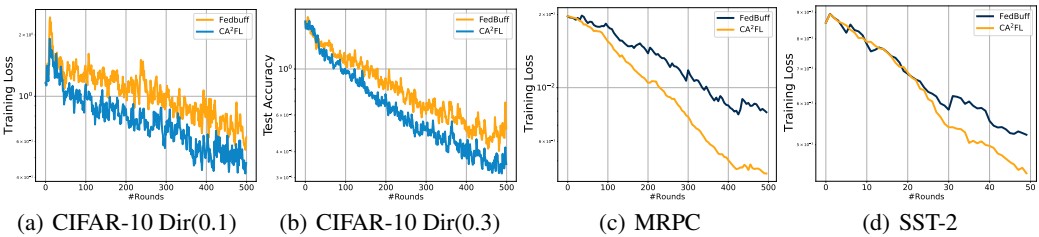

| (a) CIFAR-10 Dir(0.1) | (b) CIFAR-10 Dir(0.3) | (c) MRPC | (d) SST-2 |

Figure 1: Training/fine-tuning loss on several models and datasets.

Table 3: The result of Bert-base model on several language datasets with data heterogeneity degrees Dir(0.6). We report the mean evaluation metrics and the standard derivation for the last 5 rounds.

| Method | MRPC Acc. & std. | SST-2 Acc. & std. | RTE Acc. & std. | CoLA Acc. & std. |
|---|---|---|---|---|
| FedAsync | $\mathbf{82.86} \pm 0.42$ | $87.32 \pm 3.76$ | $62.09 \pm 0.76$ | $54.53 \pm 1.52$ |
| FedBuff | $78.68 \pm 0.41$ | $86.06 \pm 3.86$ | $60.07 \pm 1.09$ | $55.57 \pm 0.94$ |
| CA$^2$FL | $79.26 \pm 0.12$ | $\mathbf{90.76} \pm 1.02$ | $\mathbf{65.63} \pm 0.35$ | $\mathbf{56.10} \pm 0.25$ |

We also conduct a detailed comparison for studying the overall training/fine-tuning speedup for our proposed method to investigate the efficiency of our proposed method in Table 4. We simulate the wall-clock delay by assuming 80% of clients have normal local training processes, 10% have mild delays, and the last 10% have severe delays. Specifically, we observe that for each task, a client would finish the local training in $t_{\text{train}}$ seconds, then the simulated local training time for the normal clients would be $t_i \times t_{\text{train}}$, where $t_i \sim$ Uniform(0.5, 1) $\times t_{\text{train}}$, for the mild delay clients, there is $t_i \times t_{\text{train}}$, where $t_i \sim$ Uniform(1, 2) $\times t_{\text{train}}$, for severe delay clients, there is $t_i \sim$ Uniform(2, 3). We've provided an ablation study about different simulation settings in the Appendix. From Table 4 we observe

Table 2: The test accuracy of different models on the CIFAR-100 dataset with different data heterogeneity degrees. We report the mean accuracy and the standard derivation for the last 5 rounds.

| Method | Dir(0.1) Acc. & std | Dir(0.01) Acc. & std |
|---|---|---|
| FedAsync | $\mathbf{62.91} \pm 1.67$ | - |
| FedBuff | $57.12 \pm 0.60$ | $32.49 \pm 1.31$ |
| CA$^2$FL | $59.50 \pm 0.24$ | $\mathbf{37.30} \pm 0.26$ |

that our proposed CA$^2$FL maintains better training/fine-tuning efficiency among all asynchronous methods. CA$^2$FL also shows the advantage of asynchronous learning in image classification tasks. However, the efficiency of CA$^2$FL is still challenged compared to synchronized FL in the case of fine-tuning pre-trained models. We would like to take this phenomenon as a conclusion of future work for further research.

Table 4: Training/fine-tuning time simulation (in units of 10 seconds) to reach target validation accuracy (Matthew's correlation for CoLA). For each dataset, the concurrency is fixed for fair comparison. **Bold** represents the best evaluation results and the underline represents the best results for asynchronous FL.

| | Acc. | FedAsync | FedBuff | CA$^2$FL | FedAvg |
|---|---|---|---|---|---|
| CIFAR-10 | 80% | 268.80 | 291.53 | **214.16** | 388.64 |
| CIFAR-100 | 55% | 333.47 | 295.49 | **233.49** | 476.78 |
| MRPC | 80% | 2549.54 | 403.95 | **87.39** | 97.71 |
| SST-2 | 90% | 2853.5 | 2079.35 | 648.71 | **572.01** |
| RTE | 63% | 815.94 | 420.83 | **79.61** | 95.17 |
| CoLA | 55% | 217.23 | 144.64 | 34.75 | **0.79** |

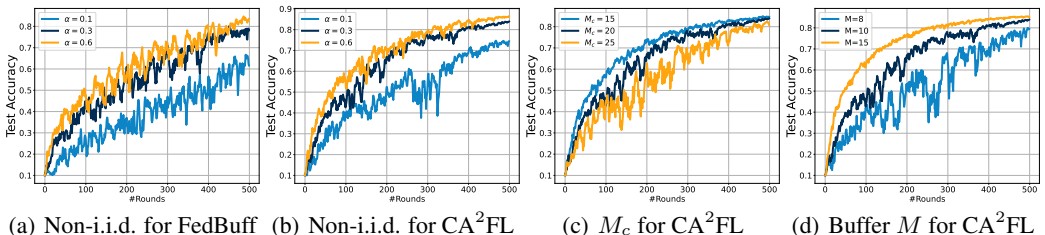

(a) Non-i.i.d. for FedBuff    (b) Non-i.i.d. for CA$^2$FL    (c) $M_c$ for CA$^2$FL    (d) Buffer $M$ for CA$^2$FL

Figure 2: Test accuracy of ablation studies for FedBuff and CA$^2$FL in training CIFAR-10 on ResNet-18 model.

We conduct ablation studies to investigate the effect of the effect of data heterogeneity, the delay simulation strategies, and the relationship between the concurrency and the buffer size $M$. Due to constraints on space, we leave detailed ablation results and discussions in Appendix A. From Figure 3 plots (a) and (b), we can observe the impact of data heterogeneity for FedBuff and CA$^2$FL. They show that the CA$^2$FL is overall less sensitive to data heterogeneity than FedBuff with less fluctuation. Plot (c) shows the the ablation study for the concurrency $M_c$ for fixed buffer $M = 10$, it shows that the accuracy decreases as the concurrency increases, with the same buffer $M = 10$. Plot (d) shows the impact of the buffer $M$ with fixed concurrency $M_c = 20$. We observe that as the increase of buffer size $M$, the overall performance increases w.r.t. the global round $T$.

## 7 CONCLUSIONS

In this paper, we first investigate the convergence of FedBuff under non-convex heterogeneous data distribution settings and we show that the data heterogeneity amplifies the negative impact of asynchronous delay which slows down the convergence of asynchronous federated learning. To address this convergence degradation issue, we propose a novel asynchronous federated learning method, CA$^2$FL, which involves caching and reusing previous updates for global calibration. We provide theoretical analysis under non-convex stochastic settings that demonstrate the significant convergence improvement of our proposed CA$^2$FL. Empirical results demonstrate the superior performance of the proposed CA$^2$FL compared to general asynchronous federated learning, and it also shows that the proposed MF-CA$^2$FL could largely save the memory overhead while maintaining the superior performance benefits from the cached update.

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

# A    ADDITIONAL EXPERIMENTS

In this section, we present additional empirical results for our proposed methods in training CNN network as in Wang & Ji (2022) on CIFAR-10, and ResNet-18 network He et al. (2016) on CIFAR-10/100 Krizhevsky et al. (2009) datasets, and fine-tuning Bert-base Devlin et al. (2018) model on GLUE datasetWang et al. (2018). Ablations and discussions about our proposed methods are also provided. All experiments in this paper are conducted on 4 NVIDIA RTX A6000 GPUs.

## A.1    MEMORY FRIENDLY CACHED-AIDED ASYNCHRONOUS FL

While CA$^2$FL successfully tackles the data heterogeneity issue in Asynchronous FL, it involves extra memory costs for maintaining the cached variable for each client on the server. However, this memory overhead can pose challenges when applying CA$^2$FL in practice, especially for large models with massive trainable parameters. To overcome this memory overhead, we extend the proposed CA$^2$FL to a memory-friendly adaption method (MF-CA$^2$FL). The main difference between CA$^2$FL and MF-CA$^2$FL lies in whether the server maintains a full-size or a quantized latest update. Specifically, in MF-CA$^2$FL, after the client $i$ obtains the model differences $\mathbf{\Delta}^i_{t-\tau^i_t}$ and sends it to the server, the server quantizes $\mathbf{\Delta}^i_{t-\tau^i_t}$ to $\mathcal{Q}(\mathbf{\Delta}^i_{t-\tau^i_t})$ via unbiased quantization approaches such as 8-bit or 4-bit quantization and keeps $\mathcal{Q}(\mathbf{\Delta}^i_{t-\tau^i_t})$ in memory. The server updates the global calibration variable $\mathbf{v}_t$ same as CA$^2$FL. Note that for each global round $t$, the server updates the quantized $\mathcal{Q}(\mathbf{\Delta}^i_{t-\tau^i_t})$ as the cached update, i.e., $\mathbf{h}^i_{t+1} = \mathcal{Q}(\mathbf{\Delta}^i_{t-\tau^i_t}), \forall i \in \mathcal{S}_t$, that being said, the cached variable $\mathbf{h}^i_{t+1}$ for each client represents the latest quantized model update difference. Therefore, compared to CA$^2$FL, this memory-friendly adaption effectively reduces the memory overhead.

## A.2    ADDITIONAL EXPERIMENTAL RESULTS

**Results on CIFAR-10.**  Table 5 shows the overall test accuracy of experiments on CIFAR-10 on training different models with two data heterogeneity levels. It demonstrates that our proposed CA$^2$FL achieve better test accuracy than asynchronous federated learning baselines. Particularly, when the data is highly heterogeneously distributed across clients, indicated by smaller $\alpha$ values in Dirichlet sampling strategies, our CA$^2$FL method significantly outperforms the other asynchronous baseline. Particularly, when $\alpha = 0.1$, CA$^2$FL can significantly outperform FedBuff with more than a 5% increase on training ResNet-18. Moreover, in the memory-friendly version MF-CA$^2$FL, which reduces the memory overhead by keeping the quantized cached update, the superior performance of the cached variable is still observed and leading to better test accuracy than asynchronous baseline. Furthermore, Figure 3 provides the test accuracy curves of training CNN and ResNet-18 networks on CIFAR-10 with $\alpha = 0.3$, offering a visual illustration of the effectiveness of our proposed method.

Table 5: The test accuracy of different models on the CIFAR-10 dataset with different models and data heterogeneity degrees. We report the mean accuracy and the standard derivationfor the last 5 rounds.

| Method | Dir(0.3) | | Dir(0.1) | |
| --- | --- | --- | --- | --- |
| | CNN Acc. & std | ResNet-18 Acc. & std | CNN Acc. & std | ResNet-18 Acc. & std |
| FedAsync | $62.29 \pm 0.16$ | $79.8 \pm 2.28$ | - | $40.58 \pm 2.92$ |
| FedBuff | $60.74 \pm 1.18$ | $78.53 \pm 3.31$ | $53.96 \pm 0.10$ | $63.03 \pm 3.17$ |
| CA$^2$FL | $\mathbf{64.40} \pm 0.32$ | $\mathbf{83.79} \pm 0.34$ | $\mathbf{57.62} \pm 0.42$ | $68.37 \pm 1.97$ |
| MF-CA$^2$FL (8 bits) | $62.43 \pm 0.04$ | $83.07 \pm 0.43$ | $57.00 \pm 0.40$ | $\mathbf{71.85} \pm 1.57$ |
| MF-CA$^2$FL (4 bits) | $62.41 \pm 0.19$ | $82.72 \pm 0.39$ | $56.20 \pm 0.54$ | $70.98 \pm 1.82$ |

**Results on CIFAR-100.**  Table 6 presents the overall test accuracy of experiments on CIFAR-100 with two data heterogeneity levels. It demonstrates that our proposed CA$^2$FL achieve higher test accuracy compared to the asynchronous federated learning baselines. Specifically, when the data is

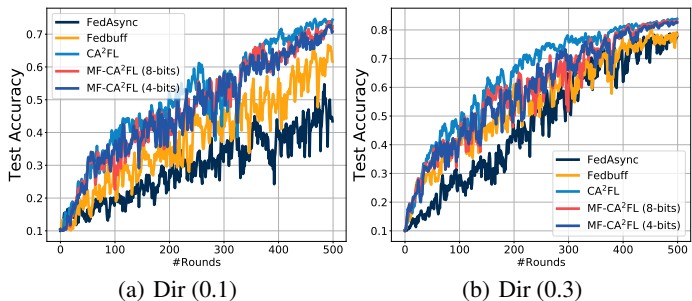

(a) Dir (0.1)    (b) Dir (0.3)

Figure 3: The test accuracy for our proposed CA$^2$FL and MF-CA$^2$FL (4 bits) with asynchronous federated learning baselines in training CIFAR10 data on ResNet-18 model.

highly heterogeneously distributed, e.g., $\alpha = 0.01$, our CA$^2$FL method significantly outperforms asynchronous baselines with more than 4.5% improvement compared to Asynchronous FL. The memory-friendly version MF-CA$^2$FL also shows its advantage over asynchronous federated learning baselines when severely data heterogeneity settings.

Table 6: The test accuracy of different models on the CIFAR-100 dataset with different data heterogeneity degrees. We report the mean accuracy and the standard derivation for the last 5 rounds.

| Method | Dir(0.1) Acc. & std | Dir(0.01) Acc. & std |
|---|---|---|
| FedAsync | **62.91** $\pm$ 1.67 | - |
| FedBuff | 57.12 $\pm$ 0.60 | 32.49 $\pm$ 1.31 |
| CA$^2$FL | 59.50 $\pm$ 0.24 | 37.30 $\pm$ 0.26 |
| MF-CA$^2$FL (8 bits) | 59.12 $\pm$ 0.21 | **37.34** $\pm$ 0.43 |
| MF-CA$^2$FL (4 bits) | 59.50 $\pm$ 0.35 | 37.29 $\pm$ 0.36 |

**Results on Tiny Imagenet.** Table 7 shows the overall test accuracy of experiments on Tiny Imagenet-200 (Le & Yang, 2015; Krizhevsky et al., 2012) on fine-tuning a pre-trained ResNet-18 (He et al., 2016) and ResNet-34 (He et al., 2016) models under non-i.i.d. data distribution settings. Similar to previous image classification tasks, there are 100 clients in total, and we set the concurrency $M_c = 20$ and update buffer $M = 10$, and we set a highly heterogeneous data distribution with Dir (0.01). Table 7 demonstrates that our proposed CA$^2$FL achieves better test accuracy than asynchronous federated learning baselines.

Table 7: The test accuracy of two models on the Tiny Imagenet dataset. We report the mean accuracy and the standard derivation for the last 5 rounds.

| Method | ResNet-18 Acc. & std | ResNet-34 Acc. & std |
|---|---|---|
| FedAsync | 50.74 $\pm$ 1.08 | 54.13 $\pm$ 1.12 |
| FedBuff | 55.55 $\pm$ 0.37 | 61.85 $\pm$ 0.38 |
| CA$^2$FL | **56.17** $\pm$ **0.23** | **62.51** $\pm$ **0.20** |

**Results on E2E NLG Challenge.** Table 8 shows the validation loss of experiments on E2E NLG Challenge (Novikova et al., 2017) on parameter-efficient fine-tuning a pre-trained GPT-2 small (Radford et al., 2019) model under non-i.i.d. data distribution settings. We set 10 clients in total, with concurrency $M_c = 5$ and update buffer $M = 2$. Since there are no labels for the generation tasks, we naturally sample a heterogeneous data distribution among clients. Similar to the previous

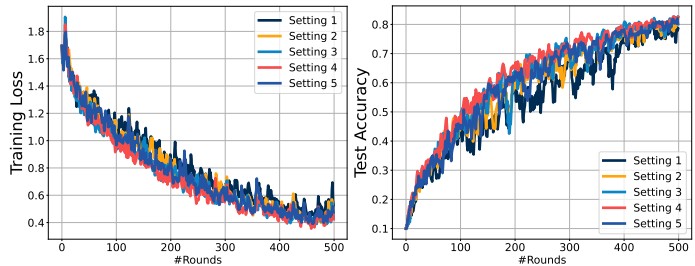

Figure 4: Ablation for several settings for wall-clock delay simulation.

language classification tasks, we adopt a LoRA fine-tuning with $\alpha_{\text{LoRA}} = 1$ and $r = 1$. Table 8 demonstrates that our proposed $\text{CA}^2\text{FL}$ achieves lower validation loss than asynchronous federated learning baselines. This further shows the effectiveness of our proposed method on various tasks.

Table 8: The validation loss of fine-tuning GPT-2 small model on E2E NLG Challenge. We report the mean loss value and the standard derivation for the last 5 rounds.

| Method | Loss & std. |
|---|---|
| FedAsync | $0.1533 \pm 0.0438$ |
| FedBuff | $0.1241 \pm 0.0110$ |
| $\text{CA}^2\text{FL}$ | $\mathbf{0.1025 \pm 0.0004}$ |

### A.2.1 ADDITIONL RESULTS

Table 9: Several settings for wall-clock delay simulation.

| | |
|---|---|
| Setting 1 | 80% from $U(0.5, 1)$, 10% from $U(1, 2)$, 10% from $U(2, 3)$ |
| Setting 2 | 80% from $U(0.5, 1)$, 10% from $U(1, 3)$, 10% from $U(3, 5)$ |
| Setting 3 | 80% from $U(0.5, 1)$, 10% from $U(1, 5)$, 10% from $U(5, 10)$ |
| Setting 4 | 80% from $U(0.5, 1)$, 20% from $U(5, 10)$ |
| Setting 5 | 60% from $U(0.5, 1)$, 40% from $U(5, 10)$ |

**Simulated delay distributions.** We simulate the delay distributions on various settings summarized in Table 9. From Figure 4, it shows that changing the setting of delay sampling would not make significant effect on both convergence and generalization.

**Performance under concept shift.** In real-world applications, the non-stationary data sources (model drift) can result in some long-term challenges. Although this is not the main focus of this paper, we conduct an experiment to examine whether our proposed method maintains its advantage under time-varying distribution shifts. We utilize a similar setting of sudden drift as in Flash (Panchal et al., 2023): all the clients suffer from the distribution change abruptly at the same round (the concept drift occurs at the 200th and 400th rounds). The concept shift is conducted as follows: for a task with $n$ labels, we swap the $i$-th label with $i + 1$-th label, $i \in [0, 1, ..., n - 1]$, and we swap the $n$-th label with label 0. The following results, demonstrate that our proposed $\text{CA}^2\text{FL}$ still achieves better performance compared to other asynchronous FL baselines when there exist distribution shifts. Since this is not our main focus in this paper, we did not have any specific design for the model drift issues but it is an interesting direction and we will leave it as our future work.

Table 10: The test accuracy training ResNet-18 model on CIFAR-10 dataset. We report the mean accuracy and the standard derivation for the last 5 rounds.

| Method | Acc. & std. |
|--------|-------------|
| FedAsync | $78.49 \pm 3.24$ |
| FedBuff | $74.17 \pm 4.61$ |
| CA$^2$FL | $\textbf{79.91} \pm \textbf{1.00}$ |

### A.3 HYPER-PARAMETERS DETAILS

**Image classifications.** We conduct detailed hyper-parameter searches to find the best hyper-parameter for each baseline. We grid over the local learning rater $\eta_l \in \{0.001, 0.01, 0.1, 1.0\}$, and the global learning rate $\eta \in \{0.1, 1.0, 2.0\}$ for each methods. Table 11 summarizes the hyper-parameter details in our experiments. Experiments are set up with 100 total clients, the concurrency is $M_c = 20$ by default, and we let the server update the global model once it receives $M = 10$ updates from clients. For each method, we conduct 2 local epochs (the explicit local iterations $K$ may differ from clients) of local training with a batch size of 50 by default. We set the weight decay as $10^{-4}$ for the local SGD optimizer. For FedAsync Xie et al. (2019), we additionally grid over the weight of the regularization term $\rho \in \{0.01, 0.1, 1.0\}$, the momentum factor $\alpha_t \in \{0.1, 0.3, 0.5, 0.9\}$.

**Languages tasks.** We conduct detailed hyper-parameter searches to find the best hyper-parameter for each baseline. We grid over the local learning rater $\eta_l \in \{5 \times 10^{-5}, 10^{-4}, 5 \times 10^{-4}, 10^{-3}\}$, and the global learning rate $\eta \in \{0.1, 1.0, 2.0\}$ for each methods. Table 11 summarizes the hyper-parameter details in our experiments. Experiments are set up with 10 total clients, the concurrency is $M_c = 5$ by default, and we let the server update the global model once it receives $M = 3$ updates from clients. For each method, we conduct 1 local epochs (the explicit local iterations $K$ may differ from clients) of local training with a batch size of 32 by default. We set $\beta_1 = 0.9, \beta_2 = 0.999, \epsilon = 10^{-6}$ and weight decay as $10^{-4}$ for the local AdamW optimizer. For FedAsync Xie et al. (2019), we additionally grid over the weight of the regularization term $\rho \in \{0.01, 0.1, 1.0\}$, the momentum factor $\alpha_t \in \{0.1, 0.3, 0.5, 0.9\}$.

Table 11: Hyper-parameters details.

| | \multicolumn CIFAR-10 | | | | | | | | | |
|---|---|---|---|---|---|---|---|---|---|---|
| | FedAsync | | FedBuff | | CA$^2$FL | | MF-CA$^2$FL (8 bits) | | MF-CA$^2$FL (4 bits) | |
| Models & Dir($\alpha$) | $\eta_l$ | $\eta$ | $\eta_l$ | $\eta$ | $\eta_l$ | $\eta$ | $\eta_l$ | $\eta$ | $\eta_l$ | $\eta$ |
| CNN & Dir(0.3) | 0.001 | 1.0 | 0.01 | 1.0 | 0.01 | 1.0 | 0.01 | 1.0 | 0.01 | 1.0 |
| ResNet-18 & Dir(0.3) | 0.01 | 1.0 | 0.01 | 1.0 | 0.01 | 1.0 | 0.01 | 1.0 | 0.01 | 1.0 |
| CNN & Dir(0.1) | - | - | 0.01 | 1.0 | 0.01 | 1.0 | 0.01 | 1.0 | 0.01 | 1.0 |
| ResNet-18 & Dir(0.1) | 0.001 | 1.0 | 0.01 | 1.0 | 0.01 | 1.0 | 0.01 | 1.0 | 0.01 | 1.0 |
| | \multicolumn CIFAR-100 | | | | | | | | | |
| | FedAsync | | FedBuff | | CA$^2$FL | | MF-CA$^2$FL (8 bits) | | MF-CA$^2$FL (4 bits) | |
| Models & Dir($\alpha$) | $\eta_l$ | $\eta$ | $\eta_l$ | $\eta$ | $\eta_l$ | $\eta$ | $\eta_l$ | $\eta$ | $\eta_l$ | $\eta$ |
| ResNet-18 & Dir(0.1) | 0.001 | 1.0 | 0.01 | 1.0 | 0.01 | 1.0 | 0.01 | 1.0 | 0.01 | 1.0 |
| ResNet-18 & Dir(0.01) | - | - | 0.01 | 1.0 | 0.01 | 1.0 | 0.01 | 1.0 | 0.01 | 1.0 |

## B FURTHER DISCUSSION ABOUT THEOREM 3.4

**Discussions.** Compare to the original proof in FedBuff (Nguyen et al., 2022), our analysis for Theorem 3.4 eliminates the unrealistic bounded gradient assumption of $\|\nabla F_i\|^2 \leq G$. Furthermore, it obtains a tighter dependency of gradient delay $\tau_t^i$. While FedBuff's original analysis has a $\tau_{\max}^2$ dependency, and we obtain a $\tau_{\max}\tau_{\text{avg}}$ dependency. Compared to FedAsync (Xie et al., 2019),

Table 12: Hyper-parameters details.

| | GLUE | | | | | |
| | FedAsync | | FedBuff | | CA$^2$FL | |
| Models & Dir($\alpha$) | $\eta_l$ | $\eta$ | $\eta_l$ | $\eta$ | $\eta_l$ | $\eta$ |
|---|---|---|---|---|---|---|
| MRPC | $5 \times 10^{-4}$ | 1.0 | $5 \times 10^{-4}$ | 1.0 | $5 \times 10^{-4}$ | 1.0 |
| SST-2 | $5 \times 10^{-4}$ | 1.0 | $5 \times 10^{-4}$ | 1.0 | $5 \times 10^{-4}$ | 1.0 |
| RTE | $5 \times 10^{-4}$ | 1.0 | $5 \times 10^{-4}$ | 1.0 | $5 \times 10^{-4}$ | 1.0 |
| CoLA | $5 \times 10^{-4}$ | 1.0 | $5 \times 10^{-4}$ | 1.0 | $5 \times 10^{-4}$ | 1.0 |

Theorem 3.4 eliminates the assumption for weak convexity, providing a more common nonconvex analysis for the asynchronous FL method.

It's worth noting that (Koloskova et al., 2022) relaxes the delay dependency in the convergence rate for distributed asynchronous SGD to $\sqrt{\tau_{\max}\tau_{\text{avg}}}$. In our study, we focus on the FL scenario where each client performs multiple local update steps prior to global aggregation. Particularly, by setting $K = 1$ and selecting a delay-dependent learning rate of $\eta = \Theta(\frac{\sqrt{M}}{\sqrt{\tau_{\max}\tau_{\text{avg}}}})$, we can achieve the same delay dependency in convergence rate of $\sqrt{\tau_{\max}\tau_{\text{avg}}}$ as in (Koloskova et al., 2022). Additionally, while (Mishchenko et al., 2022) explores the arbitrary delay in asynchronous SGD, our approach yields an improved convergence rate in heterogeneous scenarios compared to their results under similar assumptions.

**Comparisons with Toghani & Uribe (2022).** Our original FedBuff analysis in Theorem 3.4 obtains a convergence rate of $\mathcal{O}\left(\frac{(|f_1 - f_*| + \sigma^2)}{\sqrt{TKM}}\right) + \mathcal{O}\left(\frac{\sigma^2 + K\sigma_g^2}{TK}\right) + \mathcal{O}\left(\frac{\sqrt{K}\sigma_g^2}{\sqrt{TM}}\right) + \mathcal{O}\left(\frac{K\tau_{\max}\tau_{\text{avg}}\sigma_g^2 + \tau_{\max}\sigma^2}{T}\right)$.

- If we only consider the $T$ and $\tau$ related terms, we obtain a similar convergence rate of $\mathcal{O}(\frac{1}{\sqrt{T}}) + \mathcal{O}(\frac{\tau_{\max}\tau_{\text{avg}}}{T})$ compared to $\mathcal{O}(\frac{1}{\sqrt{T}}) + \mathcal{O}(\frac{\tau_{\max}^2}{T})$ in Toghani & Uribe (2022).

- If we consider the convergence rate w.r.t. $T, K, M$ and $\tau$ related terms, when $M > K$, we obtain a similar convergence rate of $\mathcal{O}(\frac{1}{\sqrt{T}}) + \mathcal{O}(\frac{K\tau_{\max}\tau_{\text{avg}}}{T})$ compared to $\mathcal{O}(\frac{1}{\sqrt{T}}) + \mathcal{O}(\frac{K\tau_{\max}^2}{T})$ in Toghani & Uribe (2022).

Moreover, the convergence rate is highly related to the learning rate choosing. For example, when adopting a different learning rate as in Theorem 3.4, our convergence rate can match the result in [1] w.r.t. $T, K, M$ and $\tau$ without any further constraints: following Equation (C.15) in Appendix, choosing $\eta = O(1)$ and $\eta_l = O(\frac{1}{\sqrt{TK}})$, then we get $\frac{1}{T}\sum_{i=1}^{T} \mathbf{E}[\|\nabla f(x_t)\|^2] = O(\frac{(f_0 - f_*)}{\sqrt{T}}) + O(\frac{\sigma^2}{\sqrt{TKM}}) + O(\frac{\sigma_g^2}{\sqrt{TM}}) + O(\frac{\tau_{\max}\tau_{avg}\sigma_g^2}{TM}) + O(\frac{\tau_{\max}\sigma^2}{TKM}) + O(\frac{\sigma^2 + K\sigma_g^2}{TK^2})$. If we look at the $T$ and delay related terms, our rate would be $\mathcal{O}(\frac{1}{\sqrt{T}}) + \mathcal{O}(\frac{\tau_{\max}\tau_{avg}}{TM})$, and this is slightly better on the non-dominant $\mathcal{O}(\frac{\tau_{\max}\tau_{avg}}{TM})$ term than the rate in Toghani & Uribe (2022).

## C CONVERGENCE ANALYSIS FOR ASYNCHRONOUS FL

*Proof of Theorem 3.4.* Since $f$ is $L$-smooth, taking conditional expectation at time $t$, we have

$$
\mathbb{E}[f(\boldsymbol{x}_{t+1})] - f(\boldsymbol{x}_t)
$$
$$
\leq \mathbb{E}[\langle \nabla f(\boldsymbol{x}_t), \boldsymbol{x}_{t+1} - \boldsymbol{x}_t \rangle] + \frac{L}{2}\mathbb{E}[\|\boldsymbol{x}_{t+1} - \boldsymbol{x}_t\|^2]
$$
$$
= \underbrace{\mathbb{E}[\langle \nabla f(\boldsymbol{x}_t)), \eta\boldsymbol{\Delta}_t \rangle]}_{I} + \underbrace{\frac{\eta^2 L}{2}\mathbb{E}[\|\boldsymbol{\Delta}_t\|^2]}_{II}. \tag{C.1}
$$

**Bounding $I$**

$$I = \mathbb{E}[\langle \nabla f(\boldsymbol{x}_t), \eta \boldsymbol{\Delta}_t \rangle]$$

$$= \frac{\eta}{M} \mathbb{E}\left[\left\langle \nabla f(\boldsymbol{x}_t), \sum_{i \in \mathcal{M}_t} \boldsymbol{\Delta}_{t-\tau_t^i}^i \right\rangle\right]$$

$$= -\frac{\eta \eta_l}{M} \mathbb{E}\left[\left\langle \nabla f(\boldsymbol{x}_t), \sum_{i \in \mathcal{M}_t} \sum_{k=0}^{K-1} \boldsymbol{g}_{t-\tau_t^i,k}^i \right\rangle\right]$$

$$= -\frac{\eta \eta_l}{M} \mathbb{E}\left[\left\langle \nabla f(\boldsymbol{x}_t), \sum_{i \in \mathcal{M}_t} \sum_{k=0}^{K-1} \nabla F_i(\boldsymbol{x}_{t-\tau_t^i,k}^i) \right\rangle\right]$$

$$= -\eta \eta_l \mathbb{E}\left[\left\langle \nabla f(\boldsymbol{x}_t), \frac{1}{N} \sum_{i=1}^{N} \sum_{k=0}^{K-1} \nabla F_i(\boldsymbol{x}_{t-\tau_t^i,k}^i) \right\rangle\right], \tag{C.2}$$

where the second and third equation holds by the update rule. The fifth one holds by the unbiasedness of stochastic gradient. By the fact of $\langle \boldsymbol{a}, \boldsymbol{b} \rangle = \frac{1}{2}[\|\boldsymbol{a}\|^2 + \|\boldsymbol{b}\|^2 - \|\boldsymbol{a} - \boldsymbol{b}\|^2]$, we have

$$-\eta \eta_l \mathbb{E}\left[\left\langle \nabla f(\boldsymbol{x}_t), \frac{1}{N} \sum_{i=1}^{N} \sum_{k=0}^{K-1} \nabla F_i(\boldsymbol{x}_{t-\tau_t^i,k}^i) \right\rangle\right]$$

$$= -\frac{\eta \eta_l K}{2} \mathbb{E}[\|\nabla f(\boldsymbol{x}_t)\|^2] - \frac{\eta \eta_l}{2K} \mathbb{E}\left[\left\|\frac{1}{N} \sum_{i=1}^{N} \sum_{k=0}^{K-1} \nabla F_i(\boldsymbol{x}_{t-\tau_t^i,k}^i)\right\|^2\right]$$

$$+ \frac{\eta \eta_l}{2} \mathbb{E}\left[\left\|\sqrt{K} \nabla f(\boldsymbol{x}_t) - \frac{1}{N\sqrt{K}} \sum_{i=1}^{N} \sum_{k=0}^{K-1} \nabla F_i(\boldsymbol{x}_{t-\tau_t^i,k}^i)\right\|^2\right]$$

$$= -\frac{\eta \eta_l K}{2} \mathbb{E}[\|\nabla f(\boldsymbol{x}_t)\|^2] - \frac{\eta \eta_l}{2K} \mathbb{E}\left[\left\|\frac{1}{N} \sum_{i=1}^{N} \sum_{k=0}^{K-1} \nabla F_i(\boldsymbol{x}_{t-\tau_t^i,k}^i)\right\|^2\right]$$

$$+ \frac{\eta \eta_l}{2} \sum_{k=0}^{K-1} \mathbb{E}\left[\left\|\frac{1}{N} \sum_{i=1}^{N} \nabla F_i(\boldsymbol{x}_t) - \frac{1}{N} \sum_{i=1}^{N} \nabla F_i(\boldsymbol{x}_{t-\tau_t^i,k}^i)\right\|^2\right], \tag{C.3}$$

for the last term, we have

$$\frac{\eta \eta_l}{2} \sum_{k=0}^{K-1} \mathbb{E}\left[\left\|\frac{1}{N} \sum_{i=1}^{N} \nabla F_i(\boldsymbol{x}_t) - \frac{1}{N} \sum_{i=1}^{N} \nabla F_i(\boldsymbol{x}_{t-\tau_t^i,k}^i)\right\|^2\right]$$

$$\leq \frac{\eta \eta_l}{2} \sum_{k=0}^{K-1} \frac{1}{N} \sum_{i=1}^{N} \mathbb{E}[\|\nabla F_i(\boldsymbol{x}_t) - \nabla F_i(\boldsymbol{x}_{t-\tau_t^i,k}^i)\|^2]$$

$$\leq \frac{\eta \eta_l}{N} \sum_{k=0}^{K-1} \sum_{i=1}^{N} \left[\mathbb{E}[\|\nabla F_i(\boldsymbol{x}_t) - \nabla F_i(\boldsymbol{x}_{t-\tau_t^i})\|^2] + \mathbb{E}[\|\nabla F_i(\boldsymbol{x}_{t-\tau_t^i}) - \nabla F_i(\boldsymbol{x}_{t-\tau_t^i,k}^i)\|^2]\right]$$

$$\leq \frac{\eta \eta_l}{N} \sum_{k=0}^{K-1} \sum_{i=1}^{N} \left[L^2 \mathbb{E}[\|\boldsymbol{x}_t - \boldsymbol{x}_{t-\tau_t^i}\|^2] + L^2 \mathbb{E}[\|\boldsymbol{x}_{t-\tau_t^i} - \boldsymbol{x}_{t-\tau_t^i,k}^i\|^2]\right]. \tag{C.4}$$

For the second term, we have

$$\mathbb{E}[\|\boldsymbol{x}_{t-\tau_t^i} - \boldsymbol{x}_{t-\tau_t^i,k}^i\|^2] = \mathbb{E}\left[\left\|\sum_{m=0}^{k-1} \eta_l \boldsymbol{g}_{t-\tau_t^i,m}^i\right\|^2\right]$$

$$\leq 5K\eta_l^2(\sigma^2 + 6K\sigma_g^2) + 30K^2\eta_l^2 \mathbb{E}[\|\nabla f(\boldsymbol{x}_{t-\tau_t^i})\|^2] \tag{C.5}$$

For the first term, we have

$$
\begin{aligned}
\mathbb{E}[\|\boldsymbol{x}_t - \boldsymbol{x}_{t-\tau_t^i}\|^2] &= \mathbb{E}\left[\left\|\sum_{s=t-\tau_t^i}^{t-1} (\boldsymbol{x}_{s+1} - \boldsymbol{x}_s)\right\|^2\right] \\
&= \mathbb{E}\left[\left\|\eta \sum_{s=t-\tau_t^i}^{t-1} \frac{1}{M} \sum_{j\in\mathcal{M}_s} \sum_{k=0}^{K-1} \eta_l \boldsymbol{g}_{s-\tau_s^j,k}^j\right\|^2\right] \\
&= \mathbb{E}\left[\left\|\eta \sum_{s=t-\tau_t^i}^{t-1} \frac{1}{M} \sum_{j\in\mathcal{M}_s} \sum_{k=0}^{K-1} \eta_l [\boldsymbol{g}_{s-\tau_s^j,k}^j - \nabla F_j(\boldsymbol{x}_{s-\tau_s^j,k}^j) + \nabla F_j(\boldsymbol{x}_{s-\tau_s^j,k}^j)]\right\|^2\right] \\
&= \mathbb{E}\left[\left\|\eta \sum_{s=t-\tau_t^i}^{t-1} \frac{1}{M} \sum_{j\in\mathcal{M}_s} \sum_{k=0}^{K-1} \eta_l [\boldsymbol{g}_{s-\tau_s^j,k}^j - \nabla F_j(\boldsymbol{x}_{s-\tau_s^j,k}^j)]\right\|^2\right] + \mathbb{E}\left[\left\|\eta \sum_{s=t-\tau_t^i}^{t-1} \frac{1}{M} \sum_{j\in\mathcal{M}_s} \sum_{k=0}^{K-1} \eta_l \nabla F_j(\boldsymbol{x}_{s-\tau_s^j,k}^j)]\right\|^2\right] \\
&\leq \frac{2\tau_{\max} K \eta^2 \eta_l^2}{M}\sigma^2 + \frac{2\tau_{\max}\eta^2\eta_l^2}{M^2} \sum_{s=t-\tau_t^i}^{t-1} \mathbb{E}\left[\left\|\sum_{j\in\mathcal{M}_s}\sum_{k=0}^{K-1}\nabla F_j(\boldsymbol{x}_{s-\tau_s^j,k}^j)\right\|^2\right], \qquad\text{(C.6)}
\end{aligned}
$$

For simplicity, we assume that the clients' participation distributions are simulated as independently uniform distribution, then for the last term, we have

$$
\begin{aligned}
&\mathbb{E}\left[\left\|\sum_{j\in\mathcal{M}_s}\sum_{k=0}^{K-1}\nabla F_j(\boldsymbol{x}_{s-\tau_s^j,k}^j)\right\|^2\right] \\
&= \frac{M(N-M)}{N(N-1)}\sum_{j=1}^{N}\mathbb{E}\left[\left\|\sum_{k=0}^{K-1}\nabla F_j(\boldsymbol{x}_{s-\tau_s^j,k}^j)\right\|^2\right] + \frac{M(M-1)}{N(N-1)}\mathbb{E}\left[\left\|\sum_{j=1}^{N}\sum_{k=0}^{K-1}\nabla F_j(\boldsymbol{x}_{s-\tau_s^j,k}^j)\right\|^2\right] \\
&\leq \frac{M(N-M)}{N(N-1)}\left[15NK^3\eta_l^2(\sigma^2+6K\sigma_g^2) + (90K^4L^2\eta_l^2+3K^2)\sum_{j=1}^{N}\mathbb{E}[\|\nabla f(\boldsymbol{x}_{s-\tau_s^j})\|^2] + 3NK^2\sigma_g^2\right] \\
&\quad + \frac{M(M-1)}{N(N-1)}\mathbb{E}\left[\left\|\sum_{j=1}^{N}\sum_{k=0}^{K-1}\nabla F_j(\boldsymbol{x}_{s-\tau_s^j,k}^j) \pm \sum_{j=1}^{N}\sum_{k=0}^{K-1}\nabla F_j(\boldsymbol{x}_{s-\tau_s^j})\right]\right] \\
&\leq \frac{M(N-M)}{N(N-1)}\left[15NK^3\eta_l^2(\sigma^2+6K\sigma_g^2) + (90K^4L^2\eta_l^2+3K^2)\sum_{j=1}^{N}\mathbb{E}[\|\nabla f(\boldsymbol{x}_{s-\tau_s^j})\|^2] + 3NK^2\sigma_g^2\right] \\
&\quad + \frac{2M(M-1)}{N(N-1)}\mathbb{E}\left[\left\|\sum_{j=1}^{N}\sum_{k=0}^{K-1}\nabla F_j(\boldsymbol{x}_{s-\tau_s^j,k}^j) - \sum_{j=1}^{N}\sum_{k=0}^{K-1}\nabla F_j(\boldsymbol{x}_{s-\tau_s^j})\right\|^2\right] + \frac{2M(M-1)K^2}{N-1}\sum_{j=1}^{N}\mathbb{E}[\|\nabla f(\boldsymbol{x}_{s-\tau_s^j})\|^2] \\
&\leq \frac{M(N-M)}{N(N-1)}\left[15NK^3\eta_l^2(\sigma^2+6K\sigma_g^2) + (90K^4L^2\eta_l^2+3K^2)\sum_{j=1}^{N}\mathbb{E}[\|\nabla f(\boldsymbol{x}_{s-\tau_s^j})\|^2] + 3NK^2\sigma_g^2\right] \\
&\quad + \frac{2M(M-1)KL^2}{N-1}\sum_{j=1}^{N}\sum_{k=0}^{K-1}\mathbb{E}[\|\boldsymbol{x}_{s-\tau_s^j,k}^j - \boldsymbol{x}_{s-\tau_s^j}\|^2] + \frac{2M(M-1)K^2}{N-1}\sum_{j=1}^{N}\mathbb{E}[\|\nabla f(\boldsymbol{x}_{s-\tau_s^j})\|^2] \\
&\leq \left[\frac{3M(N-M)}{N-1} + \frac{2NM(M-1)}{N-1}\right]\cdot\left[5K^3L^2\eta_l^2(\sigma^2+6K\sigma_g^2) + (30K^4L^2\eta_l^2+K^2)\frac{1}{N}\sum_{j=1}^{N}\mathbb{E}[\|\nabla f(\boldsymbol{x}_{s-\tau_s^j})\|^2]\right] \\
&\quad + \frac{3M(N-M)}{N-1}K^2\sigma_g^2, \qquad\text{(C.7)}
\end{aligned}
$$

where the first inequality is inspire from Lemma E.1. Then

$$
\mathbb{E}[\|\boldsymbol{x}_t - \boldsymbol{x}_{t-\tau_t^i}\|^2] \leq \frac{2\tau_{\max} K \eta^2 \eta_l^2}{M} \sigma^2 + \frac{2\tau_{\max} \eta^2 \eta_l^2}{M^2} \sum_{s=t-\tau_t^i}^{t-1} \left\{ \left[ \frac{3M(N-M)}{N-1} + \frac{2NM(M-1)}{N-1} \right] \right.
$$
$$
\left. \cdot \left[ 5K^3 L^2 \eta_l^2 (\sigma^2 + 6K\sigma_g^2) + (30K^4 L^2 \eta_l^2 + K^2)\mathbb{E}[\|\nabla f(\boldsymbol{x}_{s-\tau_s^j})\|^2] \right] + \frac{3M(N-M)}{N-1} K^2 \sigma_g^2 \right\}.
$$
$$(C.8)$$

Thus we have

$$
\frac{\eta \eta_l}{2} \sum_{k=0}^{K-1} \mathbb{E}\left[ \left\| \frac{1}{N} \sum_{j=1}^{N} \nabla F_i(\boldsymbol{x}_t) - \frac{1}{N} \sum_{j=1}^{N} \nabla F_i(\boldsymbol{x}_{t-\tau_t^i,k}^i) \right\|^2 \right]
$$
$$
\leq \eta \eta_l K L^2 \left[ 5K\eta_l^2(\sigma^2 + 6K\sigma_g^2) + 30K^2 \eta_l^2 \frac{1}{N} \sum_{i=1}^{N} \mathbb{E}[\|\nabla f(\boldsymbol{x}_{t-\tau_t^i})\|^2] \right] + 2\eta \eta_l K L^2 \frac{\tau_{\max} K \eta^2 \eta_l^2}{M} \sigma^2
$$
$$
+ 2\eta \eta_l K L^2 \frac{\tau_{\max} \eta^2 \eta_l^2}{M^2} \frac{1}{N} \sum_{i=1}^{N} \sum_{s=t-\tau_t^i}^{t-1} \left\{ \left[ \frac{3M(N-M)}{N-1} + \frac{2NM(M-1)}{N-1} \right] \right.
$$
$$
\left. \cdot \left[ 5K^3 L^2 \eta_l^2(\sigma^2 + 6K\sigma_g^2) + (30K^4 L^2 \eta_l^2 + K^2)\mathbb{E}[\|\nabla f(\boldsymbol{x}_{s-\tau_s^j})\|^2] \right] + \frac{3M(N-M)}{N-1} K^2 \sigma_g^2 \right\}
$$
$$
\leq \eta \eta_l K L^2 \left[ 5K\eta_l^2(\sigma^2 + 6K\sigma_g^2) + 30K^2 \eta_l^2 \frac{1}{N} \sum_{i=1}^{N} \mathbb{E}[\|\nabla f(\boldsymbol{x}_{t-\tau_t^i})\|^2] \right] + 2\eta \eta_l K L^2 \frac{\tau_{\max} K \eta^2 \eta_l^2}{M} \sigma^2
$$
$$
+ 2\eta \eta_l K L^2 \frac{\tau_{\max} \eta^2 \eta_l^2}{M^2} \frac{1}{N} \sum_{i=1}^{N} \sum_{s=t-\tau_t^i}^{t-1} \left\{ \left[ \frac{3M(N-M)}{N-1} + \frac{2NM(M-1)}{N-1} \right] \right.
$$
$$
\left. \cdot \left[ 5K^3 L^2 \eta_l^2(\sigma^2 + 6K\sigma_g^2) + (30K^4 L^2 \eta_l^2 + K^2)\mathbb{E}[\|\nabla f(\boldsymbol{x}_{s-\tau_s^j})\|^2] \right] \right\} + 6\eta^3 \eta_l^3 K^3 L^2 \tau_{\max} \frac{N-M}{M(N-1)} \frac{1}{N} \sum_{i=1}^{N} \tau_t^i \sigma_g^2.
$$
$$(C.9)$$

Thus for $I$, we have

$$
I \leq -\frac{\eta \eta_l K}{2} \mathbb{E}[\|\nabla f(\boldsymbol{x}_t)\|^2] - \frac{\eta \eta_l}{2K} \mathbb{E}\left[ \left\| \frac{1}{N} \sum_{i=1}^{N} \sum_{k=0}^{K-1} \nabla F_i(\boldsymbol{x}_{t-\tau_t^i,k}^i) \right\|^2 \right]
$$
$$
+ \eta \eta_l K L^2 \left[ 5K\eta_l^2(\sigma^2 + 6K\sigma_g^2) + 30K^2 \eta_l^2 \frac{1}{N} \sum_{i=1}^{N} \mathbb{E}[\|\nabla f(\boldsymbol{x}_{t-\tau_t^i})\|^2] \right] + 2\eta \eta_l K L^2 \frac{\tau_{\max} K \eta^2 \eta_l^2}{M} \sigma^2
$$
$$
+ 2\eta \eta_l K L^2 \frac{\tau_{\max} \eta^2 \eta_l^2}{M^2} \frac{1}{N} \sum_{i=1}^{N} \sum_{s=t-\tau_t^i}^{t-1} \left[ \frac{3M(N-M)}{N-1} + \frac{2NM(M-1)}{N-1} \right]
$$
$$
\cdot \left[ 5K^3 L^2 \eta_l^2(\sigma^2 + 6K\sigma_g^2) + (30K^4 L^2 \eta_l^2 + K^2)\mathbb{E}[\|\nabla f(\boldsymbol{x}_{s-\tau_s^j})\|^2] \right] + 6\eta^3 \eta_l^3 K^3 L^2 \tau_{\max} \frac{N-M}{M(N-1)} \frac{1}{N} \sum_{i=1}^{N} \tau_t^i \sigma_g^2.
$$
$$(C.10)$$

**Bounding $II$**

$$II = \frac{\eta^2 L}{2}\mathbb{E}[\|\mathbf{\Delta}_t\|^2] = \frac{\eta^2 L}{2}\mathbb{E}\left[\left\|\frac{1}{M}\sum_{i\in\mathcal{M}_t}\mathbf{\Delta}_{t-\tau_t^i}^i\right\|^2\right]$$

$$\leq \eta^2 L\left\{\frac{K\eta_l^2}{M}\sigma^2 + \frac{\eta_l^2(N-M)}{NM(N-1)}\left[15NK^3L^2\eta_l^2(\sigma^2+6K\sigma_g^2)+(90NK^4L^2\eta_l^2+3K^2)\right.\right.$$

$$\left.\cdot\sum_{i=1}^N\mathbb{E}[\|\nabla f(\boldsymbol{x}_{t-\tau_t^i})\|^2]+3NK^2\sigma_g^2\right]+\frac{\eta_l^2(M-1)}{NM(N-1)}\mathbb{E}\left[\left\|\sum_{i=1}^N\sum_{k=0}^{K-1}\nabla F_i(\boldsymbol{x}_{t-\tau_t^i,k}^i)\right\|^2\right]\right\},$$

$$(\text{C.11})$$

where the inequality holds by Lemma E.1. For simplicity, in the following, we define $\mathbf{V}_t = \sum_{i=1}^N\sum_{k=0}^{K-1}\nabla F_i(\mathbf{x}_{t-\tau_t^i,k}^i)$.

**Merging pieces.** Therefore, by merging pieces together, we have

$$\mathbb{E}[f(\mathbf{z}_{t+1})]-f(\mathbf{z}_t)=I+II$$

$$\leq -\frac{\eta\eta_l K}{2}\mathbb{E}[\|\nabla f(\boldsymbol{x}_t)\|^2]-\frac{\eta\eta_l}{2K}\mathbb{E}\left[\left\|\frac{1}{N}\sum_{i=1}^N\sum_{k=0}^{K-1}\nabla F_i(\boldsymbol{x}_{t-\tau_t^i,k}^i)\right\|^2\right]$$

$$+\eta\eta_l KL^2\left[5K\eta_l^2(\sigma^2+6K\sigma_g^2)+30K^2\eta_l^2\frac{1}{N}\sum_{i=1}^N\mathbb{E}[\|\nabla f(\boldsymbol{x}_{t-\tau_t^i})\|^2]\right]+2\eta\eta_l KL^2\frac{\tau_{\max}K\eta^2\eta_l^2}{M}\sigma^2$$

$$+2\eta\eta_l KL^2\frac{\tau_{\max}\eta^2\eta_l^2}{M^2}\frac{1}{N}\sum_{i=1}^N\sum_{s=t-\tau_t^i}^{t-1}\left[\frac{3M(N-M)}{N-1}+\frac{2NM(M-1)}{N-1}\right]$$

$$\cdot\left[5K^3L^2\eta_l^2(\sigma^2+6K\sigma_g^2)+(30K^4L^2\eta_l^2+K^2)\mathbb{E}[\|\nabla f(\boldsymbol{x}_{s-\tau_s^j})\|^2]\right]+6\eta^3\eta_l^3K^3L^2\tau_{\max}\frac{N-M}{M(N-1)}\frac{1}{N}\sum_{i=1}^N\tau_t^i\sigma_g^2$$

$$+\eta^2 L\left\{\frac{K\eta_l^2}{M}\sigma^2+\frac{\eta_l^2(N-M)}{NM(N-1)}\left[15NK^3L^2\eta_l^2(\sigma^2+6K\sigma_g^2)+(90NK^4L^2\eta_l^2+3K^2)\sum_{i=1}^N\mathbb{E}[\|\nabla f(\boldsymbol{x}_{t-\tau_t^i})\|^2]\right.\right.$$

$$\left.\left.+3NK^2\sigma_g^2\right]+\frac{\eta_l^2(M-1)}{NM(N-1)}\mathbb{E}\left[\left\|\sum_{i=1}^N\sum_{k=0}^{K-1}\nabla F_i(\mathbf{x}_{t-\tau_t^i,k}^i)\right\|^2\right]\right\}$$

$$\leq -\frac{\eta\eta_l K}{2}\mathbb{E}[\|\nabla f(\boldsymbol{x}_t)\|^2]+\eta\eta_l KL^2\left[5K\eta_l^2(\sigma^2+6K\sigma_g^2)+30K^2\eta_l^2\frac{1}{N}\sum_{i=1}^N\mathbb{E}[\|\nabla f(\boldsymbol{x}_{t-\tau_t^i})\|^2]\right]$$

$$+2\eta\eta_l KL^2\frac{\tau_{\max}K\eta^2\eta_l^2}{M}\sigma^2+2\eta\eta_l KL^2\frac{\tau_{\max}\eta^2\eta_l^2}{M^2}\frac{1}{N}\sum_{i=1}^N\sum_{s=t-\tau_t^i}^{t-1}\left[\frac{3M(N-M)}{N-1}+\frac{2NM(M-1)}{N-1}\right]$$

$$\cdot\left[5K^3L^2\eta_l^2(\sigma^2+6K\sigma_g^2)+(30K^4L^2\eta_l^2+K^2)\mathbb{E}[\|\nabla f(\boldsymbol{x}_{s-\tau_s^j})\|^2]\right]+6\eta^3\eta_l^3K^3L^2\tau_{\max}\frac{N-M}{M(N-1)}\frac{1}{N}\sum_{i=1}^N\tau_t^i\sigma_g^2$$

$$+\eta^2 L\left\{\frac{K\eta_l^2}{M}\sigma^2+\frac{\eta_l^2(N-M)}{M(N-1)}\left[15K^3L^2\eta_l^2(\sigma^2+6K\sigma_g^2)\right.\right.$$

$$\left.\left.+(90K^4L^2\eta_l^2+3K^2)\frac{1}{N}\sum_{i=1}^N\mathbb{E}[\|\nabla f(\boldsymbol{x}_{t-\tau_t^i})\|^2]+3K^2\sigma_g^2\right]\right\}+\left(\frac{\eta^2\eta_l^2 L(M-1)}{NM(N-1)}-\frac{\eta\eta_l}{2KN^2}\right)\mathbb{E}[\|\mathbf{V}_t\|^2].$$

$$(\text{C.12})$$

By organizing and merging similar terms, we have

$$
\mathbb{E}[f(\mathbf{z}_{t+1})] - f(\mathbf{z}_t)
$$

$$
\leq -\frac{\eta\eta_l K}{2}\mathbb{E}[\|\nabla f(\boldsymbol{x}_t)\|^2] + \eta\eta_l K L^2\left[5K\eta_l^2(\sigma^2 + 6K\sigma_g^2) + 30K^2\eta_l^2\frac{1}{N}\sum_{i=1}^{N}\mathbb{E}[\|\nabla f(\boldsymbol{x}_{t-\tau_t^i})\|^2]\right]
$$

$$
+ \frac{2\eta^3\eta_l^3 K^2 L^2\tau_{\max}}{M}\sigma^2 + 2\eta^3\eta_l^3 K L^2\tau_{\max}\frac{1}{MN}\sum_{i=1}^{N}\sum_{s=t-\tau_t^i}^{t-1}\left[\frac{3(N-M)}{N-1} + \frac{2N(M-1)}{N-1}\right]
$$

$$
\cdot\left[5K^3 L^2\eta_l^2(\sigma^2 + 6K\sigma_g^2) + (30K^4 L^2\eta_l^2 + K^2)\mathbb{E}[\|\nabla f(\boldsymbol{x}_{s-\tau_s^j})\|^2]\right] + 6\eta^3\eta_l^3 K^3 L^2\tau_{\max}\frac{N-M}{M(N-1)}\frac{1}{N}\sum_{i=1}^{N}\tau_t^i\sigma_g^2
$$

$$
+ \eta^2 L\left\{\frac{K\eta_l^2}{M}\sigma^2 + \frac{\eta_l^2(N-M)}{M(N-1)}\left[15K^3 L^2\eta_l^2(\sigma^2 + 6K\sigma_g^2)\right.\right.
$$

$$
\left.\left. + (90K^4 L^2\eta_l^2 + 3K^2)\frac{1}{N}\sum_{i=1}^{N}\mathbb{E}[\|\nabla f(\boldsymbol{x}_{t-\tau_t^i})\|^2] + 3K^2\sigma_g^2\right]\right\} + \left(\frac{\eta^2\eta_l^2 L(M-1)}{NM(N-1)} - \frac{\eta\eta_l}{2KN^2}\right)\mathbb{E}[\|\mathbf{V}_t\|^2].
$$

$$
\text{(C.13)}
$$

Summing over $t = 1$ to $T$, we have

$$
\mathbb{E}[f(\mathbf{z}_{T+1})] - f(\mathbf{z}_1)
$$

$$
\leq -\frac{\eta\eta_l K}{2}\sum_{t=1}^{T}\mathbb{E}[\|\nabla f(\boldsymbol{x}_t)\|^2] + \eta\eta_l K L^2\left[5K\eta_l^2 T(\sigma^2 + 6K\sigma_g^2) + 30K^2\eta_l^2\frac{1}{N}\sum_{t=1}^{T}\sum_{i=1}^{N}\mathbb{E}[\|\nabla f(\boldsymbol{x}_{t-\tau_t^i})\|^2]\right]
$$

$$
+ \frac{2\eta^3\eta_l^3 K^2 L^2\tau_{\max}T}{M}\sigma^2 + 2\eta^3\eta_l^3 K L^2\tau_{\max}\frac{1}{MN}\sum_{t=1}^{T}\sum_{i=1}^{N}\sum_{s=t-\tau_t^i}^{t-1}\left[\frac{3(N-M)}{N-1} + \frac{2N(M-1)}{N-1}\right]
$$

$$
\cdot\left[5K^3 L^2\eta_l^2(\sigma^2 + 6K\sigma_g^2) + (30K^4 L^2\eta_l^2 + K^2)\mathbb{E}[\|\nabla f(\boldsymbol{x}_{s-\tau_s^j})\|^2]\right]
$$

$$
+ 6\eta^3\eta_l^3 K^3 L^2\tau_{\max}\frac{N-M}{M(N-1)}\frac{1}{N}\sum_{t=1}^{T}\sum_{i=1}^{N}\tau_t^i\sigma_g^2 + \eta^2 L\left\{\frac{KT\eta_l^2}{M}\sigma^2 + \frac{\eta_l^2(N-M)}{M(N-1)}\left[15K^3 TL^2\eta_l^2(\sigma^2 + 6K\sigma_g^2)\right.\right.
$$

$$
\left.\left. + (90K^4 TL^2\eta_l^2 + 3K^2)\frac{1}{N}\sum_{i=1}^{N}\sum_{t=1}^{T}\mathbb{E}[\|\nabla f(\boldsymbol{x}_t)\|^2] + 3K^2 T\sigma_g^2\right]\right\} + \left(\frac{\eta^2\eta_l^2 L(M-1)}{NM(N-1)} - \frac{\eta\eta_l}{2KN^2}\right)\sum_{t=1}^{T}\mathbb{E}[\|\mathbf{V}_t\|^2]
$$

$$
\leq -\frac{\eta\eta_l K}{2}\sum_{t=1}^{T}\mathbb{E}[\|\nabla f(\boldsymbol{x}_t)\|^2] + \eta\eta_l K T L^2[5K\eta_l^2(\sigma^2 + 6K\sigma_g^2) + 30K^2\eta_l^2\tau_{\max}\sum_{t=1}^{T}\mathbb{E}[\|\nabla f(\boldsymbol{x}_t)\|^2]]
$$

$$
+ \frac{2\eta^3\eta_l^3 K^2 L^2\tau_{\max}T}{M}\sigma^2 + \frac{2\eta^3\eta_l^3 K L^2\tau_{\max}^2}{M}\sum_{t=1}^{T}\left[\frac{3(N-M)}{N-1} + \frac{2N(M-1)}{N-1}\right]
$$

$$
\cdot\left[5K^3 L^2\eta_l^2(\sigma^2 + 6K\sigma_g^2) + (30K^4 L^2\eta_l^2 + K^2)\tau_{\max}\mathbb{E}[\|\nabla f(\boldsymbol{x}_t)\|^2]\right] + 6\eta^3\eta_l^3 K^3 L^2\tau_{\max}\tau_{\text{avg}}T\frac{N-M}{M(N-1)}\sigma_g^2
$$

$$
+ \eta^2 L\left\{\frac{KT\eta_l^2}{M}\sigma^2 + \frac{\eta_l^2(N-M)}{M(N-1)}\left[15K^3 TL^2\eta_l^2(\sigma^2 + 6K\sigma_g^2)\right.\right.
$$

$$
\left.\left. + (90K^4 TL^2\eta_l^2 + 3K^2)\sum_{t=1}^{T}\mathbb{E}[\|\nabla f(\boldsymbol{x}_t)\|^2] + 3K^2 T\sigma_g^2\right]\right\} + \left(\frac{\eta^2\eta_l^2 L(M-1)}{NM(N-1)} - \frac{\eta\eta_l}{2KN^2}\right)\sum_{t=1}^{T}\mathbb{E}[\|\mathbf{V}_t\|^2].
$$

$$
\text{(C.14)}
$$

thus by specific constraint on the learning rate $\eta_l$ and $\eta$, i.e., $\eta_l \leq \frac{1}{K}$ and $\eta\eta_l K \leq \frac{1}{4\tau_{\max}^{3/2}}$, we have

$$
\begin{aligned}
\frac{1}{T}\sum_{t=1}^{T}\mathbb{E}[\|\nabla f(\boldsymbol{x}_t)\|^2] \leq {} & \frac{1}{\eta\eta_l KT}[f(\boldsymbol{x}_1) - \mathbb{E}[f(\boldsymbol{x}_{t+1})]] + L^2 5K\eta_l^2(\sigma^2 + 6K\sigma_g^2) \\
& + \frac{2K\eta^2\eta_l^2 L^2 \tau_{\max}}{M}\sigma^2 + \frac{2\eta^2\eta_l^2 L^2 \tau_{\max}^2}{M}\left[\frac{3(N-M)}{N-1} + \frac{2N(M-1)}{N-1}\right] \\
& \cdot [5K^3 L^2 \eta_l^2(\sigma^2 + 6K\sigma_g^2)] + 3\eta^2\eta_l^2 K^2 L^2 \tau_{\max}\tau_{\text{avg}}\frac{N-M}{M(N-1)}\sigma_g^2 \\
& + \frac{\eta L}{2}\left\{\frac{\eta_l}{M}\sigma^2 + \frac{\eta_l(N-M)}{M(N-1)}[15K^2 TL^2\eta_l^2(\sigma^2 + 6K\sigma_g^2) + 3K\sigma_g^2]\right\}
\end{aligned}
$$
(C.15)

By choosing $\eta = \Theta(\sqrt{KM})$ and $\eta_l = \Theta(1/\sqrt{T}K)$, we have

$$
\begin{aligned}
\frac{1}{T}\sum_{t=1}^{T}\mathbb{E}[\|\nabla f(\boldsymbol{x}_t)\|^2] = {} & \mathcal{O}\left(\frac{[(f_0 - f_*) + \sigma^2]}{\sqrt{TKM}}\right) + \mathcal{O}\left(\frac{\sigma^2 + K\sigma_g^2}{TK}\right) \\
& + \mathcal{O}\left(\frac{\sqrt{K}}{\sqrt{TM}}\sigma_g^2\right) + \mathcal{O}\left(\frac{K\tau_{\max}\tau_{\text{avg}}\sigma_g^2 + \tau_{\max}\sigma^2}{T}\right),
\end{aligned}
$$
(C.16)

where $f_* = \arg\min_{\boldsymbol{x}} f(\boldsymbol{x})$. $\qquad\square$

# D  CONVERGENCE ANALYSIS FOR CA²FL

*Proof of Theorem 5.2.* By the update scheme of Algorithm 2, we have

$$\boldsymbol{v}_t \leftarrow \boldsymbol{h}_t + \frac{1}{M}(\boldsymbol{\Delta}^i_{t-\tau^i_t} - \boldsymbol{h}^i_{t-1}) \Rightarrow \boldsymbol{v}_t = \boldsymbol{h}_{t-1} + \frac{1}{M}\sum_{i \in \mathcal{S}_t}(\boldsymbol{\Delta}^i_{t-\tau^i_t} - \boldsymbol{h}^i_{t-1}).$$

$$\boldsymbol{v}_t = \frac{1}{N}\sum_{i \notin \mathcal{S}_t}\boldsymbol{h}^i_{t-1} + \frac{1}{N}\sum_{i \in \mathcal{S}_t}\boldsymbol{h}^i_{t-1} + \frac{1}{M}\sum_{i \in \mathcal{S}_t}(\boldsymbol{\Delta}^i_{t-\tau^i_t} - \boldsymbol{h}^i_{t-1})$$

$$= \frac{1}{N}\sum_{i \notin \mathcal{S}_t}\boldsymbol{h}^i_{t-1} + \sum_{i \in \mathcal{S}_t}\left[\left(\frac{1}{N} - \frac{1}{M}\right)\boldsymbol{h}^i_{t-1} + \frac{1}{M}\boldsymbol{\Delta}^i_{t-\tau^i_t}\right] \tag{D.1}$$

Since we $\boldsymbol{h}^i_t$ represents the state update for client $i$, and $\boldsymbol{h}^i_t$ keeps unchanged if $i \notin \mathcal{S}_t$. We also have the following

$$\boldsymbol{h}_t = \boldsymbol{h}_{t-1} + \frac{1}{N}\sum_{i \in \mathcal{S}_t}(\boldsymbol{\Delta}^i_{t-\tau^i_t} - \boldsymbol{h}^i_{t-1}) = \frac{1}{N}\sum_{i \in \mathcal{S}_t}\boldsymbol{\Delta}^i_{t-\tau^i_t} + \frac{1}{N}\sum_{i \notin \mathcal{S}_t}\boldsymbol{\Delta}^i_{t-\zeta^i_t}, \tag{D.2}$$

Since $f$ is $L$-smooth, taking conditional expectation at time $t$, we have

$$\mathbb{E}[f(\boldsymbol{x}_{t+1})] - f(\boldsymbol{x}_t)$$

$$\leq \mathbb{E}[\langle \nabla f(\boldsymbol{x}_t), \boldsymbol{x}_{t+1} - \boldsymbol{x}_t\rangle] + \frac{L}{2}\mathbb{E}[\|\boldsymbol{x}_{t+1} - \boldsymbol{x}_t\|^2]$$

$$= \underbrace{\mathbb{E}[\langle \nabla f(\boldsymbol{x}_t)), \eta\boldsymbol{v}_t\rangle]}_{I} + \underbrace{\frac{\eta^2 L}{2}\mathbb{E}[\|\boldsymbol{v}_t\|^2]}_{II}. \tag{D.3}$$

**Bounding $I$**

$$I = \mathbb{E}[\langle \nabla f(\boldsymbol{x}_t), \eta \boldsymbol{v}_t \rangle]$$

$$= \mathbb{E}\left[\left\langle \nabla f(\boldsymbol{x}_t), \frac{\eta}{M} \sum_{i \in \mathcal{S}_t} \boldsymbol{\Delta}^i_{t-\tau^i_t} + \left(\frac{\eta}{N} - \frac{\eta}{M}\right) \sum_{i \in \mathcal{S}_t} \boldsymbol{h}^i_{t-1} + \frac{\eta}{N} \sum_{i \notin \mathcal{S}_t} \boldsymbol{h}^i_{t-1} \right\rangle\right]$$

$$= -\eta \eta_l \mathbb{E}\left[\left\langle \nabla f(\boldsymbol{x}_t), \frac{1}{M} \sum_{i \in \mathcal{S}_t} \sum_{k=0}^{K-1} \boldsymbol{g}^i_{t-\tau^i_t,k} + \left(\frac{1}{N} - \frac{1}{M}\right) \sum_{i \in \mathcal{S}_t} \sum_{k=0}^{K-1} \boldsymbol{g}^i_{t-\zeta^i_t,k} + \frac{1}{N} \sum_{i \notin \mathcal{S}_t} \sum_{k=0}^{K-1} \boldsymbol{g}^i_{t-\zeta^i_t,k} \right\rangle\right]$$

$$= -\eta \eta_l \mathbb{E}\left[\left\langle \nabla f(\boldsymbol{x}_t), \frac{1}{M} \sum_{i \in \mathcal{S}_t} \sum_{k=0}^{K-1} \nabla F_i(\boldsymbol{x}^i_{t-\tau^i_t,k}) + \left(\frac{1}{N} - \frac{1}{M}\right) \sum_{i \in \mathcal{S}_t} \sum_{k=0}^{K-1} \nabla F_i(\boldsymbol{x}^i_{t-\zeta^i_t,k}) \right.\right.$$

$$\left.\left. + \frac{1}{N} \sum_{i \notin \mathcal{S}_t} \sum_{k=0}^{K-1} \nabla F_i(\boldsymbol{x}^i_{t-\zeta^i_t,k}) \right\rangle\right]$$

$$= -\eta \eta_l K \mathbb{E}\left[\left\langle \nabla f(\boldsymbol{x}_t), \frac{1}{MK} \sum_{i \in \mathcal{S}_t} \sum_{k=0}^{K-1} \nabla F_i(\boldsymbol{x}^i_{t-\tau^i_t,k}) + \left(\frac{1}{NK} - \frac{1}{MK}\right) \sum_{i \in \mathcal{S}_t} \sum_{k=0}^{K-1} \nabla F_i(\boldsymbol{x}^i_{t-\zeta^i_t,k}) \right.\right.$$

$$\left.\left. + \frac{1}{NK} \sum_{i \notin \mathcal{S}_t} \sum_{k=0}^{K-1} \nabla F_i(\boldsymbol{x}^i_{t-\zeta^i_t,k}) \right\rangle\right]$$

$$= -\frac{\eta \eta_l K}{2} \mathbb{E}[\|\nabla f(\boldsymbol{x}_t)\|^2]$$

$$- \frac{\eta \eta_l}{2K} \mathbb{E}\left[\left\| \sum_{i \in \mathcal{S}_t} \sum_{k=0}^{K-1} \left(\frac{1}{M} \nabla F_i(\boldsymbol{x}^i_{t-\tau^i_t,k}) + \left(\frac{1}{N} - \frac{1}{M}\right) \nabla F_i(\boldsymbol{x}^i_{t-\zeta^i_t,k})\right) + \frac{1}{N} \sum_{i \notin \mathcal{S}_t} \sum_{k=0}^{K-1} \nabla F_i(\boldsymbol{x}^i_{t-\zeta^i_t,k}) \right\|^2\right]$$

$$+ \frac{\eta \eta_l K}{2} \mathbb{E}\left[\left\| \nabla f(\boldsymbol{x}_t) - \frac{1}{K}\left[ \sum_{i \in \mathcal{S}_t} \sum_{k=0}^{K-1} \left(\frac{1}{M} \nabla F_i(\boldsymbol{x}^i_{t-\tau^i_t,k}) + \left(\frac{1}{N} - \frac{1}{M}\right) \nabla F_i(\boldsymbol{x}^i_{t-\zeta^i_t,k})\right) \right.\right.\right.$$

$$\left.\left.\left. + \frac{1}{N} \sum_{i \notin \mathcal{S}_t} \sum_{k=0}^{K-1} \nabla F_i(\boldsymbol{x}^i_{t-\zeta^i_t,k}) \right] \right\|^2\right], \tag{D.4}$$

where the second and third equation holds by the update rule. The forth one holds by the unbiasedness of stochastic gradient, and the last one holds by the fact of $\langle \boldsymbol{a}, \boldsymbol{b} \rangle = \frac{1}{2}[\|\boldsymbol{a}\|^2 + \|\boldsymbol{b}\|^2 - \|\boldsymbol{a} - \boldsymbol{b}\|^2]$. For the last item, we have

$$\frac{\eta \eta_l K}{2} \mathbb{E}\left[\left\| \nabla f(\boldsymbol{x}_t) - \frac{1}{K}\left[ \sum_{i \in \mathcal{S}_t} \sum_{k=0}^{K-1} \left(\frac{1}{M} \nabla F_i(\boldsymbol{x}^i_{t-\tau^i_t,k}) + \left(\frac{1}{N} - \frac{1}{M}\right) \nabla F_i(\boldsymbol{x}^i_{t-\zeta^i_t,k})\right) \right.\right.\right.$$

$$\left.\left.\left. + \sum_{i \notin \mathcal{S}_t} \sum_{k=0}^{K-1} \frac{1}{N} \nabla F_i(\boldsymbol{x}^i_{t-\zeta^i_t,k}) \right] \right\|^2\right]$$

$$= \frac{\eta \eta_l K}{2} \mathbb{E}\left[\left\| \frac{1}{NK} \sum_{i=1}^{N} \sum_{k=0}^{K-1} \nabla F_i(\boldsymbol{x}_t) - \frac{1}{K}\left[ \sum_{i \in \mathcal{S}_t} \sum_{k=0}^{K-1} \frac{1}{M} \nabla F_i(\boldsymbol{x}^i_{t-\tau^i_t,k}) + \left(\frac{1}{N} - \frac{1}{M}\right) \nabla F_i(\boldsymbol{x}^i_{t-\zeta^i_t,k}) \right.\right.\right.$$

$$\left.\left.\left. + \sum_{i \notin \mathcal{S}_t} \sum_{k=0}^{K-1} \frac{1}{N} \nabla F_i(\boldsymbol{x}^i_{t-\zeta^i_t,k}) \right] \right\|^2\right] = (*), \tag{D.5}$$

then

$$
\begin{aligned}
(*) =& \frac{\eta\eta_l K}{2}\mathbb{E}\Bigg[\Bigg\|\frac{1}{MK}\sum_{i\in\mathcal{S}_t}\sum_{k=0}^{K-1}[\nabla F_i(\boldsymbol{x}_{t-\tau_t^i})-\nabla F_i(\boldsymbol{x}_{t-\tau_t^i,k}^i)]+\left(\frac{1}{N}-\frac{1}{M}\right)\frac{1}{K}\sum_{i\in\mathcal{S}_t}\sum_{k=0}^{K-1}[\nabla F_i\boldsymbol{x}_{t-\zeta_t^i})-\nabla F_i(\boldsymbol{x}_{t-\zeta_t^i,k}^i)] \\
&+\frac{1}{NK}\sum_{i\notin\mathcal{S}_t}\sum_{k=0}^{K-1}[\nabla F_i(\boldsymbol{x}_{t-\zeta_t^i})-\nabla F_i(\boldsymbol{x}_{t-\zeta_t^i,k}^i)]+\frac{1}{M}\sum_{i\in\mathcal{S}_t}[\nabla F_i(\boldsymbol{x}_t)-\nabla F_i(\boldsymbol{x}_{t-\tau_t^i})] \\
&+\left(\frac{1}{N}-\frac{1}{M}\right)\sum_{i\in\mathcal{S}_t}[\nabla F_i(\boldsymbol{x}_t)-\nabla F_i(\boldsymbol{x}_{t-\zeta_t^i})]+\frac{1}{N}\sum_{i\notin\mathcal{S}_t}[\nabla F_i(\boldsymbol{x}_t)-\nabla F_i(\boldsymbol{x}_{t-\zeta_t^i})]\Bigg\|^2\Bigg] \\
\leq & \eta\eta_l K\mathbb{E}\Bigg[\Bigg\|\frac{1}{MK}\sum_{i\in\mathcal{S}_t}\sum_{k=0}^{K-1}[\nabla F_i(\boldsymbol{x}_{t-\tau_t^i})-\nabla F_i(\boldsymbol{x}_{t-\tau_t^i,k}^i)]+\left(\frac{1}{N}-\frac{1}{M}\right)\frac{1}{K}\sum_{i\in\mathcal{S}_t}\sum_{k=0}^{K-1}[\nabla F_i(\boldsymbol{x}_{t-\zeta_t^i})-\nabla F_i(\boldsymbol{x}_{t-\zeta_t^i,k}^i)] \\
&+\frac{1}{NK}\sum_{i\notin\mathcal{S}_t}\sum_{k=0}^{K-1}[\nabla F_i(\boldsymbol{x}_{t-\zeta_t^i})-\nabla F_i(\boldsymbol{x}_{t-\zeta_t^i,k}^i)]\Bigg\|^2\Bigg]+\eta\eta_l K\mathbb{E}\Bigg[\Bigg\|\frac{1}{M}\sum_{i\in\mathcal{S}_t}[\nabla F_i(\boldsymbol{x}_t)-\nabla F_i(\boldsymbol{x}_{t-\tau_t^i})] \\
&+\left(\frac{1}{N}-\frac{1}{M}\right)\sum_{i\in\mathcal{S}_t}[\nabla F_i(\boldsymbol{x}_t)-\nabla F_i(\boldsymbol{x}_{t-\zeta_t^i})]+\frac{1}{N}\sum_{i\notin\mathcal{S}_t}[\nabla F_i(\boldsymbol{x}_t)-\nabla F_i(\boldsymbol{x}_{t-\zeta_t^i})]\Bigg\|^2\Bigg],
\end{aligned}
$$

$$(\text{D.6})$$

where we have

$$
\begin{aligned}
&\eta\eta_l K\mathbb{E}\Bigg[\Bigg\|\frac{1}{MK}\sum_{i\in\mathcal{S}_t}\sum_{k=0}^{K-1}[\nabla F_i(\boldsymbol{x}_{t-\tau_t^i})-\nabla F_i(\boldsymbol{x}_{t-\tau_t^i,k}^i)]+\left(\frac{1}{N}-\frac{1}{M}\right)\frac{1}{K}\sum_{i\in\mathcal{S}_t}\sum_{k=0}^{K-1}[\nabla F_i(\boldsymbol{x}_{t-\zeta_t^i})-\nabla F_i(\boldsymbol{x}_{t-\zeta_t^i,k}^i)] \\
&+\frac{1}{NK}\sum_{i\notin\mathcal{S}_t}\sum_{k=0}^{K-1}[\nabla F_i(\boldsymbol{x}_{t-\zeta_t^i})-\nabla F_i(\boldsymbol{x}_{t-\zeta_t^i,k}^i)]\Bigg\|^2\Bigg] \\
\leq & \frac{3\eta\eta_l K}{M}\mathbb{E}\Bigg[\sum_{i\in\mathcal{S}_t}\Bigg\|\frac{1}{K}\sum_{k=0}^{K-1}[\nabla F_i(\boldsymbol{x}_{t-\tau_t^i})-\nabla F_i(\boldsymbol{x}_{t-\tau_t^i,k}^i)]\Bigg\|^2\Bigg] \\
&+\frac{3\eta\eta_l K(N-M)^2}{N^2M}\mathbb{E}\Bigg[\sum_{i\in\mathcal{S}_t}\Bigg\|\frac{1}{K}\sum_{k=0}^{K-1}[\nabla F_i(\boldsymbol{x}_{t-\zeta_t^i})-\nabla F_i(\boldsymbol{x}_{t-\zeta_t^i,k}^i)]\Bigg\|^2\Bigg] \\
&+\frac{3\eta\eta_l K(N-M)}{N^2}\mathbb{E}\Bigg[\sum_{i\notin\mathcal{S}_t}\Bigg\|\frac{1}{K}\sum_{k=0}^{K-1}[\nabla F_i(\boldsymbol{x}_{t-\zeta_t^i})-\nabla F_i(\boldsymbol{x}_{t-\zeta_t^i,k}^i)]\Bigg\|^2\Bigg] \\
\leq & \frac{3\eta\eta_l K}{M}\sum_{i\in\mathcal{S}_t}[5KL^2\eta_l^2(\sigma^2+6K\sigma_g^2)+30K^2L^2\eta_l^2\mathbb{E}[\|\nabla f(\boldsymbol{x}_{t-\tau_t^i})\|^2]] \\
&+\frac{3\eta\eta_l K(N-M)^2}{N^2M}\sum_{i\in\mathcal{S}_t}[5KL^2\eta_l^2(\sigma^2+6K\sigma_g^2)+30K^2L^2\eta_l^2\mathbb{E}[\|\nabla f(\boldsymbol{x}_{t-\zeta_t^i})\|^2]] \\
&+\frac{3\eta\eta_l K(N-M)}{N^2}\sum_{i\notin\mathcal{S}_t}[5KL^2\eta_l^2(\sigma^2+6K\sigma_g^2)+30K^2L^2\eta_l^2\mathbb{E}[\|\nabla f(\boldsymbol{x}_{t-\zeta_t^i})\|^2]] \\
\leq & 3\eta\eta_l K\cdot[5KL^2\eta_l^2(\sigma^2+6K\sigma_g^2)]+\frac{90\eta\eta_l^3 K^3 L^2}{M}\sum_{i\in\mathcal{S}_t}\mathbb{E}[\|\nabla f(\boldsymbol{x}_{t-\tau_t^i})\|^2] \\
&+\frac{6\eta\eta_l K(N-M)^2}{N^2}[5KL^2\eta_l^2(\sigma^2+6K\sigma_g^2)]+\frac{90\eta\eta_l^3 K^3 L^2(N-M)^2}{N^2M}\sum_{i\in\mathcal{S}_t}\mathbb{E}[\|\nabla f(\boldsymbol{x}_{t-\zeta_t^i})\|^2] \\
&+\frac{90\eta\eta_l^3 K^3 L^2(N-M)}{N^2}\sum_{i\notin\mathcal{S}_t}\mathbb{E}[\|\nabla f(\boldsymbol{x}_{t-\zeta_t^i})\|^2]]
\end{aligned}
$$

$$(\text{D.7})$$

We also have

$$
\eta\eta_l K \mathbb{E}\left[\left\|\frac{1}{M}\sum_{i\in\mathcal{S}_t}[\nabla F_i(\boldsymbol{x}_t)-\nabla F_i(\boldsymbol{x}_{t-\tau_t^i})]+\left(\frac{1}{N}-\frac{1}{M}\right)\sum_{i\in\mathcal{S}_t}[\nabla F_i(\boldsymbol{x}_t)-\nabla F_i(\boldsymbol{x}_{t-\zeta_t^i})]\right.\right.
$$

$$
\left.\left.+\frac{1}{N}\sum_{i\notin\mathcal{S}_t}[\nabla F_i(\boldsymbol{x}_t)-\nabla F_i(\boldsymbol{x}_{t-\zeta_t^i})]\right\|^2\right]
$$

$$
\leq\frac{3\eta\eta_l K}{M}\mathbb{E}\left[\sum_{i\in\mathcal{S}_t}\|\nabla F_i(\boldsymbol{x}_t)-\nabla F_i(\boldsymbol{x}_{t-\tau_t^i})\|^2\right]+\frac{3\eta\eta_l K(N-M)^2}{N^2 M}\mathbb{E}\left[\sum_{i\in\mathcal{S}_t}\|\nabla F_i(\boldsymbol{x}_t)-\nabla F_i(\boldsymbol{x}_{t-\zeta_t^i})\|^2\right]
$$

$$
+\frac{3\eta\eta_l K(N-M)}{N^2}\mathbb{E}\left[\sum_{i\notin\mathcal{S}_t}\|\nabla F_i(\boldsymbol{x}_t)-\nabla F_i(\boldsymbol{x}_{t-\zeta_t^i})\|^2\right]
$$

$$
\leq\frac{3\eta\eta_l K L^2}{M}\mathbb{E}\left[\sum_{i\in\mathcal{S}_t}\left\|\sum_{s=t-\tau_t^i}^{t-1}(\boldsymbol{x}_{s+1}-\boldsymbol{x}_s)\right\|^2\right]+\frac{3\eta\eta_l K(N-M)^2 L^2}{N^2 M}\mathbb{E}\left[\sum_{i\in\mathcal{S}_t}\left\|\sum_{s=t-\zeta_t^i}^{t-1}(\boldsymbol{x}_{s+1}-\boldsymbol{x}_s)\right\|^2\right]
$$

$$
+\frac{3\eta\eta_l K(N-M)L^2}{N^2}\mathbb{E}\left[\sum_{i\notin\mathcal{S}_t}\left\|\sum_{s=t-\zeta_t^i}^{t-1}(\boldsymbol{x}_{s+1}-\boldsymbol{x}_s)\right\|^2\right]. \tag{D.8}
$$

Then we have

$$
\frac{3\eta\eta_l K L^2}{M}\mathbb{E}\left[\sum_{i\in\mathcal{S}_t}\left\|\sum_{s=t-\tau_t^i}^{t-1}(\boldsymbol{x}_{s+1}-\boldsymbol{x}_s)\right\|^2\right]+\frac{3\eta\eta_l K(N-M)^2 L^2}{N^2 M}\mathbb{E}\left[\sum_{i\in\mathcal{S}_t}\left\|\sum_{s=t-\zeta_t^i}^{t-1}(\boldsymbol{x}_{s+1}-\boldsymbol{x}_s)\right\|^2\right]
$$

$$
+\frac{3\eta\eta_l K(N-M)L^2}{N^2}\mathbb{E}\left[\sum_{i\notin\mathcal{S}_t}\left\|\sum_{s=t-\zeta_t^i}^{t-1}(\boldsymbol{x}_{s+1}-\boldsymbol{x}_s)\right\|^2\right]
$$

$$
=\frac{3\eta\eta_l K L^2}{N}\mathbb{E}\left[\sum_{i=1}^{N}\left\|\sum_{s=t-\tau_t^i}^{t-1}(\boldsymbol{x}_{s+1}-\boldsymbol{x}_s)\right\|^2\right]+\frac{3\eta\eta_l K(N-M)^2 L^2}{N^3}\mathbb{E}\left[\sum_{i=1}^{N}\left\|\sum_{s=t-\zeta_t^i}^{t-1}(\boldsymbol{x}_{s+1}-\boldsymbol{x}_s)\right\|^2\right]
$$

$$
+\frac{3\eta\eta_l K(N-M)^2 L^2}{N^3}\mathbb{E}\left[\sum_{i=1}^{N}\left\|\sum_{s=t-\zeta_t^i}^{t-1}(\boldsymbol{x}_{s+1}-\boldsymbol{x}_s)\right\|^2\right]
$$

$$
=\frac{3\eta\eta_l K L^2}{N}\mathbb{E}\left[\sum_{i=1}^{N}\left\|\sum_{s=t-\tau_t^i}^{t-1}(\boldsymbol{x}_{s+1}-\boldsymbol{x}_s)\right\|^2\right]+\frac{6\eta\eta_l K(N-M)^2 L^2}{N^3}\mathbb{E}\left[\sum_{i=1}^{N}\left\|\sum_{s=t-\zeta_t^i}^{t-1}(\boldsymbol{x}_{s+1}-\boldsymbol{x}_s)\right\|^2\right]. \tag{D.9}
$$

Similar to the proof in the previous section, we have

$$
\mathbb{E}\left[\left\|\sum_{s=t-\tau_t^i}^{t-1}(\boldsymbol{x}_{s+1}-\boldsymbol{x}_s)\right\|^2\right]
$$

$$
=\mathbb{E}[\|\boldsymbol{x}_t-\boldsymbol{x}_{t-\tau_t^i}\|^2]
$$

$$
\leq\frac{2\tau_{\max}K\eta^2\eta_l^2}{M}\sigma^2+\frac{2\tau_{\max}\eta^2\eta_l^2}{M^2}\sum_{s=t-\tau_t^i}^{t-1}\mathbb{E}\left[\left\|\sum_{j\in\mathcal{S}_s}\sum_{k=0}^{K-1}\left(\frac{1}{M}\nabla F_i(\boldsymbol{x}_{s-\tau_s^j,k}^i)+\left(\frac{1}{N}-\frac{1}{M}\right)\nabla F_i(\boldsymbol{x}_{s-\zeta_s^i,k}^i)\right)\right.\right.
$$

$$
\left.\left.+\frac{1}{N}\sum_{j\notin\mathcal{S}_s}\sum_{k=0}^{K-1}\nabla F_i(\boldsymbol{x}_{s-\zeta_s^i,k}^i)\right\|^2\right], \tag{D.10}
$$

and

$$
\mathbb{E}\left[\left\|\sum_{s=t-\zeta_t^i}^{t-1}(\boldsymbol{x}_{s+1}-\boldsymbol{x}_s)\right\|^2\right]
$$

$$
= \mathbb{E}[\|\boldsymbol{x}_t - \boldsymbol{x}_{t-\zeta_t^i}\|^2]
$$

$$
\leq \frac{2\zeta_{\max}K\eta^2\eta_l^2}{M}\sigma^2 + \frac{2\zeta_{\max}\eta^2\eta_l^2}{M^2}\sum_{s=t-\zeta_t^i}^{t-1}\mathbb{E}\left[\left\|\sum_{j\in\mathcal{S}_s}\sum_{k=0}^{K-1}\left(\frac{1}{M}\nabla F_i(\boldsymbol{x}_{s-\tau_s^j,k}^i) + \left(\frac{1}{N}-\frac{1}{M}\right)\nabla F_i(\boldsymbol{x}_{s-\zeta_s^i,k}^i)\right)\right.\right.
$$

$$
\left.\left. + \frac{1}{N}\sum_{j\notin\mathcal{S}_s}\sum_{k=0}^{K-1}\nabla F_i(\boldsymbol{x}_{s-\zeta_s^i,k}^i)\right\|^2\right], \tag{D.11}
$$

**Bounding $II$**

$$
II = \frac{\eta^2 L}{2}\mathbb{E}[\|\boldsymbol{v}_t\|^2] = \frac{\eta^2 L}{2}\mathbb{E}\left[\left\|\frac{1}{M}\sum_{i\in\mathcal{S}_t}\boldsymbol{\Delta}_{t-\tau_t^i}^i + \left(\frac{1}{N}-\frac{1}{M}\right)\sum_{i\in\mathcal{S}_t}\boldsymbol{h}_t^i + \frac{1}{N}\sum_{i\notin\mathcal{S}_t}\boldsymbol{h}_t^i\right\|^2\right]
$$

$$
\leq \eta^2\eta_l^2 L\mathbb{E}\left[\left\|\sum_{i\in\mathcal{S}_t}\sum_{k=0}^{K-1}\left(\frac{1}{M}[\boldsymbol{g}_{t-\tau_t^i,k}^i - \nabla F_i(\boldsymbol{x}_{t-\tau_t^i,k}^i)] + \left(\frac{1}{N}-\frac{1}{M}\right)[\boldsymbol{g}_{t-\zeta_t^i,k}^i - \nabla F_i(\boldsymbol{x}_{t-\zeta_t^i,k}^i)]\right)\right.\right.
$$

$$
\left.\left. + \frac{1}{N}\sum_{i\notin\mathcal{S}_t}\sum_{k=0}^{K-1}[\boldsymbol{g}_{t-\zeta_t^i,k}^i - \nabla F_i(\boldsymbol{x}_{t-\zeta_t^i,k}^i)]\right\|^2\right] + \eta^2\eta_l^2 L\mathbb{E}\left[\left\|\sum_{i\in\mathcal{S}_t}\sum_{k=0}^{K-1}\left(\frac{1}{M}\nabla F_i(\boldsymbol{x}_{t-\tau_t^i,k}^i)\right.\right.\right.
$$

$$
\left.\left.\left. + \left(\frac{1}{N}-\frac{1}{M}\right)\nabla F_i(\boldsymbol{x}_{t-\zeta_t^i,k}^i)\right) + \frac{1}{N}\sum_{i\notin\mathcal{S}_t}\sum_{k=0}^{K-1}\nabla F_i(\boldsymbol{x}_{t-\zeta_t^i,k}^i)\right\|^2\right]
$$

$$
\leq \eta^2\eta_l^2 L\frac{1}{M^2}\sum_{i\in\mathcal{S}_t}\sum_{k=0}^{K-1}\mathbb{E}[\|\boldsymbol{g}_{t-\tau_t^i,k}^i - \nabla F_i(\boldsymbol{x}_{t-\tau_t^i,k}^i)\|^2] + \left(\frac{1}{N}-\frac{1}{M}\right)^2\sum_{i\in\mathcal{S}_t}\sum_{k=0}^{K-1}\mathbb{E}[\|\boldsymbol{g}_{t-\zeta_t^i,k}^i - \nabla F_i(\boldsymbol{x}_{t-\zeta_t^i,k}^i)\|^2]
$$

$$
+ \frac{1}{N^2}\sum_{i\notin\mathcal{S}_t}\sum_{k=0}^{K-1}\mathbb{E}[\|\boldsymbol{g}_{t-\zeta_t^i,k}^i - \nabla F_i(\boldsymbol{x}_{t-\zeta_t^i,k}^i)\|^2] + \eta^2\eta_l^2 L\mathbb{E}\left[\left\|\sum_{i\in\mathcal{S}_t}\sum_{k=0}^{K-1}\left(\frac{1}{M}\nabla F_i(\boldsymbol{x}_{t-\tau_t^i,k}^i)\right.\right.\right.
$$

$$
\left.\left.\left. + \left(\frac{1}{N}-\frac{1}{M}\right)\nabla F_i(\boldsymbol{x}_{t-\zeta_t^i,k}^i)\right) + \frac{1}{N}\sum_{i\notin\mathcal{S}_t}\sum_{k=0}^{K-1}\nabla F_i(\boldsymbol{x}_{t-\zeta_t^i,k}^i)\right\|^2\right]
$$

$$
\leq \eta^2\eta_l^2 L\frac{3K}{M}\sigma^2 + \eta^2\eta_l^2 L\mathbb{E}\left[\left\|\sum_{i\in\mathcal{S}_t}\sum_{k=0}^{K-1}\left(\frac{1}{M}\nabla F_i(\boldsymbol{x}_{t-\tau_t^i,k}^i) + \left(\frac{1}{N}-\frac{1}{M}\right)\nabla F_i(\boldsymbol{x}_{t-\zeta_t^i,k}^i)\right)\right.\right.
$$

$$
\left.\left. + \frac{1}{N}\sum_{i\notin\mathcal{S}_t}\sum_{k=0}^{K-1}\nabla F_i(\boldsymbol{x}_{t-\zeta_t^i,k}^i)\right\|^2\right]. \tag{D.12}
$$

**Merging pieces.** For simplicity, we define $\mathbf{V}_t = \sum_{i\in\mathcal{S}_t}\sum_{k=0}^{K-1}\left(\frac{1}{M}\nabla F_i(\boldsymbol{x}_{t-\tau_t^i,k}^i) + \left(\frac{1}{N} - \frac{1}{M}\right)\nabla F_i(\boldsymbol{x}_{t-\zeta_t^i,k}^i)\right) + \frac{1}{N}\sum_{i\notin\mathcal{S}_t}\sum_{k=0}^{K-1}\nabla F_i(\boldsymbol{x}_{t-\zeta_t^i,k}^i)$ Therefore, by merging pieces together, we

have

$$\mathbb{E}[f(\boldsymbol{x}_{t+1})] - f(\boldsymbol{x}_t) = I + II$$

$$\leq -\frac{\eta\eta_l K}{2}\mathbb{E}[\|\nabla f(\boldsymbol{x}_t)\|^2] - \frac{\eta\eta_l}{2K}\mathbb{E}[\|\mathbf{V}_t\|^2] + \left(3\eta\eta_l K + \frac{6\eta\eta_l K(N-M)^2}{N^2}\right)5KL^2\eta_l^2(\sigma^2 + 6K\sigma_g^2)$$

$$+ \frac{3\eta\eta_l K}{M}\sum_{i\in\mathcal{S}_t}30K^2L^2\eta_l^2\mathbb{E}[\|\nabla f(\boldsymbol{x}_{t-\tau_t^i})\|^2] + \frac{90\eta\eta_l^3 K^3 L^2(N-M)^2}{N^2 M}\sum_{i\in\mathcal{S}_t}\mathbb{E}[\|\nabla f(\boldsymbol{x}_{t-\zeta_t^i})\|^2]$$

$$+ \frac{90\eta\eta_l^3 K^3 L^2(N-M)}{N^2}\sum_{i\notin\mathcal{S}_t}\mathbb{E}[\|\nabla f(\boldsymbol{x}_{t-\zeta_t^i})\|^2] + \frac{6\tau_{\max}K\eta^2\eta_l^2 L^2}{M}\sigma^2$$

$$+ \frac{6\tau_{\max}\eta^2\eta_l^2 L^2}{M^2}\sum_{s=t-\tau_t^i}^{t-1}\mathbb{E}\left[\left\|\sum_{j\in\mathcal{S}_s}\sum_{k=0}^{K-1}\left(\frac{1}{M}\nabla F_i(\boldsymbol{x}_{s-\tau_s^j,k}^i) + \left(\frac{1}{N} - \frac{1}{M}\right)\nabla F_i(\boldsymbol{x}_{s-\zeta_s^i,k}^i)\right)\right.\right.$$

$$+ \frac{1}{N}\sum_{j\notin\mathcal{S}_s}\sum_{k=0}^{K-1}\nabla F_i(\boldsymbol{x}_{s-\zeta_s^i,k}^i)\bigg\|^2\bigg] + \frac{12\zeta_{\max}K\eta^2\eta_l^2 L^2}{M}\sigma^2 + \frac{12\zeta_{\max}\eta^2\eta_l^2 L^2}{M^2}\sum_{s=t-\zeta_t^i}^{t-1}$$

$$\cdot\mathbb{E}\left[\left\|\sum_{j\in\mathcal{S}_s}\sum_{k=0}^{K-1}\left(\frac{1}{M}\nabla F_i(\boldsymbol{x}_{s-\tau_s^j,k}^i) + \left(\frac{1}{N} - \frac{1}{M}\right)\nabla F_i(\boldsymbol{x}_{s-\zeta_s^i,k}^i)\right) + \frac{1}{N}\sum_{j\notin\mathcal{S}_s}\sum_{k=0}^{K-1}\nabla F_i(\boldsymbol{x}_{s-\zeta_s^i,k}^i)\right\|^2\right]$$

$$+ \eta^2 L\left\{\frac{3K\eta_l^2}{M}\sigma^2 + \eta_l^2\mathbb{E}[\|\mathbf{V}_t\|^2]\right\}$$

$$= -\frac{\eta\eta_l K}{2}\mathbb{E}[\|\nabla f(\boldsymbol{x}_t)\|^2] + \left(3 + \frac{6(N-M)^2}{N^2}\right)5\eta\eta_l K^2 L^2\eta_l^2(\sigma^2 + 6K\sigma_g^2) + \eta^2 L\frac{3K\eta_l^2}{M}\sigma^2$$

$$+ \frac{6\tau_{\max}K\eta^2\eta_l^2 L^2}{M}\sigma^2 + \frac{6\tau_{\max}\eta^2\eta_l^2 L^2}{M^2}\sum_{s=t-\tau_t^i}^{t-1}\mathbb{E}[\|\mathbf{V}_s\|^2] + \frac{12\zeta_{\max}K\eta^2\eta_l^2 L^2}{M}\sigma^2 + \frac{12\zeta_{\max}\eta^2\eta_l^2 L^2}{M^2}\sum_{s=t-\zeta_t^i}^{t-1}\mathbb{E}[\|\mathbf{V}_s\|^2]$$

$$+ \frac{90\eta\eta_l^3 K^3 L^2}{M}\sum_{i\in\mathcal{S}_t}\mathbb{E}[\|\nabla f(\boldsymbol{x}_{t-\tau_t^i})\|^2] + \frac{90\eta\eta_l^3 K^3 L^2(N-M)^2}{N^2 M}\sum_{i\in\mathcal{S}_t}\mathbb{E}[\|\nabla f(\boldsymbol{x}_{t-\zeta_t^i})\|^2]$$

$$+ \frac{90\eta\eta_l^3 K^3 L^2(N-M)}{N^2}\sum_{i\notin\mathcal{S}_t}\mathbb{E}[\|\nabla f(\boldsymbol{x}_{t-\zeta_t^i})\|^2] - \left(\frac{\eta\eta_l}{2K} - \eta^2\eta_l^2 L\right)\mathbb{E}[\|\mathbf{V}_t\|^2]. \tag{D.13}$$

Summing over $t = 1$ to $T$, we have

$$\mathbb{E}[f(\boldsymbol{x}_{T+1})] - f(\boldsymbol{x}_1)$$

$$\leq -\frac{\eta\eta_l K}{2}\sum_{t=1}^{T}\mathbb{E}[\|\nabla f(\boldsymbol{x}_t)\|^2] + \left(3 + \frac{6(N-M)^2}{N^2}\right)5\eta\eta_l K^2 T L^2\eta_l^2(\sigma^2 + 6K\sigma_g^2) + \eta^2 L\frac{3KT\eta_l^2}{M}\sigma^2$$

$$+ \frac{6\tau_{\max}\eta^2\eta_l^2 L^2 KT}{M}\sigma^2 + \frac{12\zeta_{\max}\eta^2\eta_l^2 L^2 KT}{M}\sigma^2$$

$$+ \frac{90\eta\eta_l^3 K^3 L^2}{M}\sum_{t=1}^{T}\sum_{i\in\mathcal{S}_t}\mathbb{E}[\|\nabla f(\boldsymbol{x}_{t-\tau_t^i})\|^2] + \frac{90\eta\eta_l^3 K^3 L^2(N-M)^2}{N^2 M}\sum_{t=1}^{T}\sum_{i\in\mathcal{S}_t}\mathbb{E}[\|\nabla f(\boldsymbol{x}_{t-\zeta_t^i})\|^2]$$

$$+ \frac{90\eta\eta_l^3 K^3 L^2(N-M)}{N^2}\sum_{t=1}^{T}\sum_{i\notin\mathcal{S}_t}\mathbb{E}[\|\nabla f(\boldsymbol{x}_{t-\zeta_t^i})\|^2]$$

$$- \left(\frac{\eta\eta_l}{2K} - \eta^2\eta_l^2 L - \frac{6\tau_{\max}^2\eta^2\eta_l^2 L^2}{M} - \frac{12\zeta_{\max}^2\eta^2\eta_l^2 L^2}{M}\right)\sum_{t=1}^{T}\mathbb{E}[\|\mathbf{V}_t\|^2], \tag{D.14}$$

with some conditions on the learning rate $\eta_l$ and $\eta$, i.e., $\eta_l \leq \frac{1}{K}$, $\frac{\eta\eta_l K}{M} \leq \frac{1}{36\tau_{\max}^2 L^2}$, $\frac{\eta\eta_l K}{M} \leq \frac{1}{72\zeta_{\max}^2 L^2}$, $\eta_l K \leq \frac{1}{36\sqrt{\tau_{\max}}L}$ and $\eta_l K \leq \frac{1}{36\sqrt{\zeta_{\max}}L}$. Therefore,

$$
\mathbb{E}[f(\boldsymbol{x}_{T+1})] - f(\boldsymbol{x}_1)
$$

$$
\leq -\frac{\eta\eta_l K}{4} \sum_{t=1}^{T} \mathbb{E}[\|\nabla f(\boldsymbol{x}_t)\|^2] + \left(3 + \frac{6(N-M)^2}{N^2}\right) 5\eta\eta_l K^2 T L^2 \eta_l^2 (\sigma^2 + 6K\sigma_g^2) + \eta^2 L \frac{3KT\eta_l^2}{M}\sigma^2
$$

$$
+ \frac{6\tau_{\max}\eta^2\eta_l^2 L^2 KT}{M}\sigma^2 + \frac{12\zeta_{\max}\eta^2\eta_l^2 L^2 KT}{M}\sigma^2. \tag{D.15}
$$

Therefore,

$$
\frac{\eta\eta_l K}{4} \sum_{t=1}^{T} \mathbb{E}[\|\nabla f(\boldsymbol{x}_t)\|^2] \leq f(\boldsymbol{x}_1) - \mathbb{E}[f(\boldsymbol{x}_{T+1})] + \left(3 + \frac{6(N-M)^2}{N^2}\right) 5\eta\eta_l K^2 T L^2 \eta_l^2 (\sigma^2 + 6K\sigma_g^2)
$$

$$
+ \eta^2 L \frac{3KT\eta_l^2}{M}\sigma^2 + \frac{6\tau_{\max}\eta^2\eta_l^2 L^2 KT}{M}\sigma^2 + \frac{12\zeta_{\max}\eta^2\eta_l^2 L^2 KT}{M}\sigma^2,
$$

$$
\frac{1}{T} \sum_{t=1}^{T} \mathbb{E}[\|\nabla f(\boldsymbol{x}_t)\|^2] \leq \frac{1}{\eta\eta_l KT}[f(\boldsymbol{x}_1) - \mathbb{E}[f(\boldsymbol{x}_{T+1})]] + \left(3 + \frac{6(N-M)^2}{N^2}\right) 5KL^2 \eta_l^2 (\sigma^2 + 6K\sigma_g^2)
$$

$$
+ \frac{3\eta\eta_l L}{M}\sigma^2 + (\tau_{\max} + 2\zeta_{\max})\frac{6\eta\eta_l KL^2\sigma^2}{M}. \tag{D.16}
$$

Hence by choosing $\eta_l = \frac{1}{\sqrt{T}K}$ and $\eta = \sqrt{KM}$, then the convergence rate satisfies

$$
\frac{1}{T} \sum_{t=1}^{T} \mathbb{E}[\|\nabla f(\boldsymbol{x}_t)\|^2] = \mathcal{O}\left(\frac{f_0 - f_*}{\sqrt{TKM}}\right) + \mathcal{O}\left(\frac{\sigma^2}{\sqrt{TKM}}\right) + \mathcal{O}\left(\frac{\sigma^2 + K\sigma_g^2}{TK}\right)
$$

$$
+ \mathcal{O}\left(\frac{(\tau_{\max} + \zeta_{\max})\sigma^2}{T}\right), \tag{D.17}
$$

where $f_* = \arg\min_{\boldsymbol{x}} f(\boldsymbol{x})$. $\qquad\square$

## E    SUPPORTING LEMMAS

**Lemma E.1.** The model difference $\boldsymbol{\Delta}_t = \frac{1}{M}\sum_{i\in\mathcal{M}_t}\boldsymbol{\Delta}_t^i = \frac{1}{M}\sum_{i\in\mathcal{M}_t}\sum_{k=0}^{K-1}\boldsymbol{g}_t^i$ with $|\mathcal{M}_t| = M$ satisfies

$$
\mathbb{E}[\|\boldsymbol{\Delta}_t\|^2] \leq \frac{2K\eta_l^2}{M}\sigma_l^2 + \frac{2\eta_l^2(N-M)}{NM(N-1)}[15NK^3L^3\eta_l^2(\sigma_l^2 + 6K\sigma_g^2) + (90NK^4L^2\eta_l^2 + 3NK^2)\|\nabla f(\boldsymbol{x}_t)\|^2
$$

$$
+ 3NK^2\sigma_g^2] + \frac{2\eta_l^2(M-1)}{NM(N-1)}\mathbb{E}\left[\left\|\sum_{i=1}^{N}\sum_{k=0}^{K-1}\nabla F_i(\boldsymbol{x}_{t,k}^i)\right\|^2\right].
$$

*Proof.* We have

$$\mathbb{E}[\|\boldsymbol{\Delta}_t\|^2] = \mathbb{E}\left[\left\|\frac{1}{M}\sum_{i\in\mathcal{M}_t}\boldsymbol{\Delta}_t^i\right\|^2\right]$$

$$= \frac{1}{M^2}\mathbb{E}\left[\left\|\sum_{i=1}^N \mathbb{I}\{i\in\mathcal{M}_t\}\boldsymbol{\Delta}_t^i\right\|^2\right]$$

$$\leq \frac{2\eta_l^2}{M^2}\mathbb{E}\left[\left\|\sum_{i=1}^N \mathbb{I}\{i\in\mathcal{M}_t\}\sum_{k=0}^{K-1}[\boldsymbol{g}_{t,k}^i - \nabla F_i(\boldsymbol{x}_{t,k}^i)]\right\|^2 + \left\|\sum_{i=1}^N \mathbb{I}\{i\in\mathcal{M}_t\}\sum_{k=0}^{K-1}\nabla F_i(\boldsymbol{x}_{t,k}^i)\right\|^2\right]$$

$$= \frac{2\eta_l^2}{M^2}\mathbb{E}\left[\left\|\sum_{i=1}^N \mathbb{I}\{i\in\mathcal{M}_t\}\sum_{k=0}^{K-1}[\boldsymbol{g}_{t,k}^i - \nabla F_i(\boldsymbol{x}_{t,k}^i)]\right\|^2 + \left\|\sum_{i=1}^N \mathbb{I}\{i\in\mathcal{M}_t\}\sum_{k=0}^{K-1}\nabla F_i(\boldsymbol{x}_{t,k}^i)\right\|^2\right]$$

$$= \frac{2\eta_l^2}{NM}\mathbb{E}\left[\left\|\sum_{i=1}^N\sum_{k=0}^{K-1}[\boldsymbol{g}_{t,k}^i - \nabla F_i(\boldsymbol{x}_{t,k}^i)]\right\|^2\right] + \frac{2\eta_l^2}{M^2}\mathbb{E}\left[\left\|\sum_{i=1}^N \mathbb{I}\{i\in\mathcal{M}_t\}\sum_{k=0}^{K-1}\nabla F_i(\boldsymbol{x}_{t,k}^i)\right\|^2\right]$$

$$\leq \frac{2K\eta_l^2}{n}\sigma_l^2 + \frac{2\eta_l^2}{M^2}\mathbb{E}\left[\left\|\sum_{i=1}^N \mathbb{I}\{i\in\mathcal{M}_t\}\sum_{k=0}^{K-1}\nabla F_i(\boldsymbol{x}_{t,k}^i)\right\|^2\right], \tag{E.1}$$

where the fifth equation holds due to $\mathbb{P}\{i\in\mathcal{M}_t\} = \frac{M}{N}$. Note that we have

$$\left\|\sum_{i=1}^N\sum_{k=0}^{K-1}\nabla F_i(\boldsymbol{x}_{t,k}^i)\right\|^2 = \sum_{i=1}^N\left\|\sum_{k=0}^{K-1}\nabla F_i(\boldsymbol{x}_{t,k}^i)\right\|^2 + \sum_{i\neq j}\left\langle\sum_{k=0}^{K-1}\nabla F_i(\boldsymbol{x}_{t,k}^i), \sum_{k=0}^{K-1}\nabla F_j(\boldsymbol{x}_{t,k}^j)\right\rangle$$

$$= \sum_{i=1}^N N\left\|\sum_{k=0}^{K-1}\nabla F_i(\boldsymbol{x}_{t,k}^i)\right\|^2 - \frac{1}{2}\sum_{i\neq j}\left\|\sum_{k=0}^{K-1}\nabla F_i(\boldsymbol{x}_{t,k}^i) - \sum_{k=0}^{K-1}\nabla F_j(\boldsymbol{x}_{t,k}^j)\right\|^2, \tag{E.2}$$

where the second equation holds due to $\|\sum_{i=1}^N \boldsymbol{x}_i\|^2 = \sum_{i=1}^N N\|\boldsymbol{x}_i\|^2 - \frac{1}{2}\sum_{i\neq j}\|\boldsymbol{x}_i - \boldsymbol{x}_j\|^2$. By the sampling strategy (without replacement), we have $\mathbb{P}\{i\in\mathcal{M}_t\} = \frac{M}{N}$ and $\mathbb{P}\{i,j\in\mathcal{M}_t\} = \frac{M(M-1)}{N(N-1)}$, thus we have

$$\left\|\sum_{i=1}^N\sum_{k=0}^{K-1}\mathbb{P}\{i\in\mathcal{M}_t\}\nabla F_i(\boldsymbol{x}_{t,k}^i)\right\|^2$$

$$= \sum_{i=1}^N \mathbb{P}\{i\in\mathcal{M}_t\}\left\|\sum_{k=0}^{K-1}\nabla F_i(\boldsymbol{x}_{t,k}^i)\right\|^2 + \sum_{i\neq j}\mathbb{P}\{i,j\in\mathcal{M}_t\}\left\langle\sum_{k=0}^{K-1}\nabla F_i(\boldsymbol{x}_{t,k}^i), \sum_{k=0}^{K-1}\nabla F_j(\boldsymbol{x}_{t,k}^j)\right\rangle$$

$$= \frac{M}{N}\sum_{i=1}^N\left\|\sum_{k=0}^{K-1}\nabla F_i(\boldsymbol{x}_{t,k}^i)\right\|^2 + \frac{M(M-1)}{N(N-1)}\sum_{i\neq j}\left\langle\sum_{k=0}^{K-1}\nabla F_i(\boldsymbol{x}_{t,k}^i), \sum_{k=0}^{K-1}\nabla F_j(\boldsymbol{x}_{t,k}^j)\right\rangle$$

$$= \frac{M^2}{N}\sum_{i=1}^N\left\|\sum_{k=0}^{K-1}\nabla F_i(\boldsymbol{x}_{t,k}^i)\right\|^2 - \frac{M(M-1)}{2N(N-1)}\sum_{i\neq j}\left\|\sum_{k=0}^{K-1}\nabla F_i(\boldsymbol{x}_{t,k}^i) - \sum_{k=0}^{K-1}\nabla F_j(\boldsymbol{x}_{t,k}^j)\right\|^2$$

$$= \frac{M(N-M)}{N(N-1)}\sum_{i=1}^N\left\|\sum_{k=0}^{K-1}\nabla F_i(\boldsymbol{x}_{t,k}^i)\right\|^2 + \frac{M(M-1)}{N(N-1)}\left\|\sum_{i=1}^N\sum_{k=0}^{K-1}\nabla F_i(\boldsymbol{x}_{t,k}^i)\right\|^2, \tag{E.3}$$

where the third equation holds due to $\langle\boldsymbol{x}, \mathbf{y}\rangle = \frac{1}{2}[\|\boldsymbol{x}\|^2 + \|\mathbf{y}\|^2 - \|\boldsymbol{x} - \mathbf{y}\|^2]$ and the last equation holds due to $\frac{1}{2}\sum_{i\neq j}\|\boldsymbol{x}_i - \boldsymbol{x}_j\|^2 = \sum_{i=1}^N N\|\boldsymbol{x}_i\|^2 - \|\sum_{i=1}^N \boldsymbol{x}_i\|^2$. Therefore, for the last term in equation E.1, we have

$$\mathbb{E}[\|\boldsymbol{\Delta}_t\|^2] \leq \frac{2K\eta_l^2}{M}\sigma_l^2 + \frac{2\eta_l^2(N-M)}{NM(N-1)}\sum_{i=1}^N\mathbb{E}\left[\left\|\sum_{k=0}^{K-1}\nabla F_i(\boldsymbol{x}_{t,k}^i)\right\|^2\right] + \frac{2\eta_l^2(M-1)}{NM(N-1)}\mathbb{E}\left[\left\|\sum_{i=1}^N\sum_{k=0}^{K-1}\nabla F_i(\boldsymbol{x}_{t,k}^i)\right\|^2\right]. \tag{E.4}$$

The second term in equation E.4 is bounded partially following Reddi et al. (2021),

$$
\begin{aligned}
\sum_{i=1}^{N}\left\|\sum_{k=0}^{K-1}\nabla F_i(\boldsymbol{x}_{t,k}^i)\right\|^2 &= \sum_{i=1}^{N}\mathbb{E}\left\|\sum_{k=0}^{K-1}[\nabla F_i(\boldsymbol{x}_{t,k}^i) - \nabla F_i(\boldsymbol{x}_t) + \nabla F_i(\boldsymbol{x}_t) - \nabla f(\boldsymbol{x}_t) + \nabla f(\boldsymbol{x}_t)]\right\|^2 \\
&\leq 3\sum_{i=1}^{N}\mathbb{E}\left\|\sum_{k=0}^{K-1}[\nabla F_i(\boldsymbol{x}_{t,k}^i) - \nabla F_i(\boldsymbol{x}_t)]\right\|^2 + 3NK^2\sigma_g^2 + 3NK^2\|\nabla f(\boldsymbol{x}_t)\|^2 \\
&\leq 3KL^2\sum_{i=1}^{N}\sum_{k=0}^{K-1}\mathbb{E}[\|\boldsymbol{x}_{t,k}^i - \boldsymbol{x}_t\|^2] + 3NK^2\sigma_g^2 + 3NK^2\|\nabla f(\boldsymbol{x}_t)\|^2 \\
&\leq 15NK^3L^3\eta_l^2(\sigma_l^2 + 6K\sigma_g^2) + (90NK^4L^2\eta_l^2 + 3NK^2)\|\nabla f(\boldsymbol{x}_t)\|^2 + 3NK^2\sigma_g^2,
\end{aligned}
\tag{E.5}
$$

where the last inequality holds by applying Lemma E.2 (also follows from Reddi et al. (2021)). Substituting equation E.5 into equation E.4, this concludes the proof. □

**Lemma E.2.** (This lemma directly follows from Lemma 3 in FedAdam Reddi et al. (2021). For local learning rate which satisfying $\eta_l \leq \frac{1}{8KL}$, the local model difference after $k$ ($\forall k \in \{0, 1, ..., K-1\}$) steps local updates satisfies

$$
\frac{1}{N}\sum_{i=1}^{N}\mathbb{E}[\|\boldsymbol{x}_{t,k}^i - \boldsymbol{x}_t\|^2] \leq 5K\eta_l^2(\sigma_l^2 + 6K\sigma_g^2) + 30K^2\eta_l^2\mathbb{E}[\|\nabla f(\boldsymbol{x}_t)\|^2].
\tag{E.6}
$$

*Proof.* The proof of Lemma E.2 is exactly same as the proof of Lemma 3 in Reddi et al. (2021). □

