# OpenReview forum: "Tackling the Data Heterogeneity in Asynchronous Federated Learning with Cached Update Calibration"
_ICLR.cc/2024/Conference — ICLR 2024 poster_

### Official Review · Reviewer_xHG4 · 2023-10-29

**Soundness:** 3 good
**Presentation:** 2 fair
**Contribution:** 3 good
**Rating:** 6
**Confidence:** 3

**Summary:**

This paper presents a significant advancement in the field of federated learning by introducing the Caching and Asynchronous Aggregation Federated Learning (CA2FL) method, a novel approach designed to optimize the learning process in non-convex stochastic settings. Recognizing the challenges posed by asynchronous updates and non-IID data in decentralized learning systems, the authors propose an innovative technique that caches and reuses previous updates for global calibration, aiming to enhance convergence and overall efficiency.

The core contribution of the research lies in its robust theoretical analysis, which underscores the method's substantial improvements in convergence. In addition to the theory, the paper provides empirical results that demonstrate CA2FL's superior performance compared to traditional asynchronous federated learning approaches. These results are particularly good, indicating that the method not only addresses key challenges in the field but also outperforms existing solutions.

Furthermore, the research introduces an iteration of the method, MF-CA2FL, designed to significantly reduce memory overhead, a critical consideration in practical applications. This variant maintains the performance benefits of the original method, indicating the approach's flexibility and adaptability.

**Strengths:**

1. By implementing a caching mechanism that reuses previous updates, the CA2FL  method optimizes the learning process, particularly in environments with non-IID data and asynchronous updates.
2. The authors don’t just stop at theoretical claims; they substantiate their findings with empirical data, demonstrating the method's superior performance in real-world scenarios, including several vision and language tasks.
3. By addressing the critical issue of memory overhead in federated learning systems, MF-CA2FL maintains the advantages of the CA2FL method while significantly reducing resource consumption, making it highly beneficial for practical deployments, especially in resource-constrained environments.

**Weaknesses:**

1.  It's unclear if the proposed CA2FL method was tested across a diverse range of scenarios, network architectures, or data distributions. This limited scope might raise questions about the method's generalizability and effectiveness in various real-world applications outside the tested environments.
2. The paper might not fully address how this complexity impacts the scalability of the method, especially in larger, more diverse decentralized systems. If the method imposes significant computational or communicational overhead, it could limit its scalability and practical adoption. For example, it is only tested on CIFAR-10/100 dataset with ResNet-18 which is a shallow small NN architecture.

**Questions:**

1.  The CA2FL method introduces a unique approach to handling asynchronous updates in federated learning, but how does it adapt to various neural network architectures and models? Given the diverse range of models used in different sectors, can the authors elaborate on the method's compatibility and performance enhancements across these varied architectures?
2.  Federated learning often operates under constraints like limited bandwidth, varied data quality, or privacy regulations. How does the CA2FL method, with its caching and asynchronous aggregation, perform under such practical constraints?
3. The paper highlights the immediate improvements in convergence and efficiency, but federated learning models often operate over extended periods, during which they may encounter non-stationary data sources (model drift). How does the CA2FL approach anticipate and handle such long-term challenges? Are there mechanisms within the method to adapt to evolving data characteristics while maintaining its efficiency and accuracy over time?

---

> ### Author Response · Authors · 2023-11-18
> **Response to Reviewer xHG4 (Part 1)**
>
> Thank you for the constructive comments.
>
> ### **W1** The diverse range of scenarios, network architectures, or data distributions when evaluated.
>
> In the original submission, we evaluated our proposed method under two vision datasets: CIFAR-10 and CIFAR-100, with two network architectures, traditional CNN and ResNet-18, and fine-tuning the Bert-base model using four language datasets from the GLUE benchmark. We provided ablations of different levels of heterogeneous data distributions and different levels of delay (most ablation studies are presented in Figure 2 and Figure 4). Moreover, during the rebuttal, we added several experiments under various settings:
>
> 1) TinyImagenet & ResNet-18 (w.r.t. test accuracy):
>
> |FedAsync |FedBuff | CA$^2$FL (propsoed)|
> |--|--|--|
> |50.74 $\pm$ 1.08 |55.55 $\pm$ 0.37 |**56.17 $\pm$ 0.23**|
>
> 2) TinyImagenet & ResNet-34 (w.r.t. test accuracy):
>
> |FedAsync |FedBuff | CA$^2$FL (propsoed)|
> |--|--|--|
> | 54.13 $\pm$ 1.12 |61.85 $\pm$ 0.38 |**62.51 $\pm$ 0.20**|
>
> 3) E2E NLG Challenge & GPT-2 small (w.r.t. validation loss):
>
> |FedAsync |FedBuff | CA$^2$FL (propsoed)|
> |--|--|--|
> |0.1533 $\pm$ 0.0438 | 0.1241 $\pm$ 0.0110|**0.1025  $\pm$ 0.0004**|
>
> The three tables above show that CA$^2$FL achieve higher accuracy than FedBuff and FedAsync on more complicated classification tasks, and obtain lower validation loss on language generation task. Therefore, we believe that sufficient experimental results can demonstrate the proposed CA$^2$FL's generalizability and effectiveness in various settings.
>
>
> ### **W2** The complexity and scalability of the proposed method, especially the computational or communicational overhead.
> First, the method we propose does not impose additional communication overhead compared to FedAvg and FedBuff. The extra computational overhead compared to FedAvg and FedBuff is also minimal since the extra computations are just numerically aggregating cached variables.
>
> Second, it is not true that we "only tested on CIFAR-10/100 dataset with ResNet-18". In our original submission, we also provided experiments on fine-tuning BERT models on several GLUE datasets. In addition, as we mentioned in the previous **W1**, during the rebuttal process, we added two more experiments for the CV task and the language task. Therefore, we believe that our experimental results can verify the effectiveness and scalability of our proposed method.

---

> ### Author Response · Authors · 2023-11-18
> **Response to Reviewer xHG4 (Part 2)**
>
> ### **Q1** The method's compatibility and performance enhancements across these varied architectures.
> As we demonstrated in responding **W1** and **W2**, we are confident our empirical results can adequately demonstrate the method's capabilities and performance enhancement across various architectures.
>
>
> ### **Q2** How does the CA2FL method, with its caching and asynchronous aggregation, perform under practical constraints in federated learning?
>
> * *Limited bandwidth*: Typically, limited bandwidth results in delays in communication. However, the introduction of asynchronous aggregation does not exacerbate communication delays. Instead, an asynchronous approach provides a more flexible updating framework, capable of adjusting to different bandwidth levels, thereby enhancing communication efficiency. Therefore, the proposed CA$^2$FL method inherits the advantage of asynchronous aggregation and could overcome some challenges with limited bandwidth.
> * *Varied data quality*: In this paper, we do not focus on the quality of each data point, but we consider the heterogeneity of data distributions among clients. As we claimed in the paper, our proposed CA$^2$FL can mitigate the performance degradation caused by data heterogeneity among clients. We will leave the data quality evaluation as future work.
> * *Privacy regulations*: As we claimed in the paper, our proposed method would not raise any additional privacy concerns (same as the traditional synchronous and asynchronous federated learning methods). Moreover, CA$^2$FL is compatible with existing privacy-preserving methods such as secure aggregation and differential privacy. Since our paper focuses on improving convergence under data heterogeneity and asynchronous delay, we will address improvements related to privacy regulations in future works.
>
> ### **Q3** How can CA$^2$FL anticipate and handle long-term model drift challenges? Are there mechanisms within the method to adapt to evolving data characteristics while maintaining its efficiency and accuracy over time?
>
> Thank you for pointing this out. The non-stationary data sources (model drift) can result in some long-term challenges, but that is not our main focus in this paper. Nevertheless, we conduct an experiment to examine whether our proposed method maintains its advantage under time-varying distribution shifts. We utilize a similar setting of sudden drift as in [1]: all the clients suffer from the distribution change abruptly at the same round (the concept drift occurs at the 200th and 400th rounds). The concept shift is conducted as follows: for a task with $n$ labels, we swap $i$-th label with $i+1$-th label, $i \in [0,1,...,n-1]$, and we swap the $n$-th label with label 0. The following results, demonstrate that our proposed CA$^2$FL still achieves better performance compared to other asynchronous FL baselines when there exist distribution shifts.
>
> |FedAsync |FedBuff | CA$^2$FL (propsoed)|
> |--|--|--|
> | 78.49 $\pm$ 3.24 |74.17 $\pm$ 4.61 |**79.91 $\pm$ 1.00**|
>
> [1] Panchal, K., Choudhary, S., Mitra, S., Mukherjee, K., Sarkhel, S., Mitra, S. &amp; Guan, H. .(2023). Flash: Concept Drift Adaptation in Federated Learning.
>
> Finally, since this is not our main focus in this paper, we did not have any specific design for the model drift issues but it is an interesting direction and we will leave it as our future work.

---

> > ### Comment · Reviewer_xHG4 · 2023-11-19
> >
> > Thank you for your response on adding additional experiments and explanations to my concerns. After checking all the responses and considering that I already gave a positive score. I will keep my current score.

---

> > > ### Author Response · Authors · 2023-11-19
> > > **Thanks for your review and response**
> > >
> > > We are glad to hear that our response has addressed your concerns. Thank you for your constructive comments and we are happy to further discuss with you any remaining questions.

---

### Official Review · Reviewer_yUBQ · 2023-10-31

**Soundness:** 3 good
**Presentation:** 4 excellent
**Contribution:** 3 good
**Rating:** 8
**Confidence:** 4

**Summary:**

Paper introduces the Cache-Aided Asynchronous Federated Learning (CA$^2$FL) algorithm which allows a central server to cache each received update from a device and use it to correct the global update. Standard convergence rate guarantees are provided, and nice empirical results showcase the improved performance of CA$^2$FL. CA$^2$FL is closely tied to FedBuff, with a small tweak to allow models to be cached and reused during training.

**Strengths:**

1. Paper is well-written, concise, and straightforward.
2. Very strong empirical results! The section is nicely done, and the algorithm is compared to relevant baselines (one other algorithm which may be quite related could also be compared against and I've mentioned it in the Questions section).

**Weaknesses:**

1. Reproducing the empirical results of FedBuff is important to motivate the Cache-Aided algorithm, but should not take up much space and shouldn't be considered a contribution.
2. The cache idea seems to stem from SAGA, and while its inclusion is non-trivial, this is the only new piece to the FedBuff algorithm and thus the contribution seems to be marginal.
3. Maximum gradient delay bound is used, however newer works no longer require this assumption [R1]. I ask below in the questions, but slight confusion with state delay and why gradient delay can't be used for it as well.

[R1] Koloskova, Anastasiia, Sebastian U. Stich, and Martin Jaggi. "Sharper convergence guarantees for asynchronous sgd for distributed and federated learning." Advances in Neural Information Processing Systems 35 (2022): 17202-17215.

**Questions:**

1. Remark 3.5 and the overall convergence analysis of FedBuff is not very novel. Equation (4) in FedBuff also details the $\tau_{max}$ and $\sigma^2$ issues. What is novel by reproducing it?
2. The state delay seems closely tied to the gradient delay. Why can't the state delay take the form of the gradient delay? The inclusion seems to muddle the convergence result in Equation (5.1).
3. Do the convergence rates align with FedBuff and FedAsync? Are they better or worse? A table would be nice to detail this comparison.
4. How much memory is added to the training process when models are cached? I could see that this would be problematic in real-world implementations with thousands or hundreds of thousands of devices participating in training.
5. The idea of caching seems quite similar to the recent ICLR work [R2], except in the decentralized setting. [R2] seems to also utilize some sense of caching and it might be a closer comparison in the asynchronous FL domain.

I am interested to see how to authors respond to my questions and concerns. I feel that this paper is very strong empirically, which is important. At the same time, the novelty of the proposed method does not seem to clear the bar of an accepted paper.

[R2] Bornstein, Marco and Rabbani, Tahseen and Wang, Evan and Bedi, Amrit Singh and Huang, Furong. "SWIFT: Rapid Decentralized Federated Learning via Wait-Free Model Communication." International Conference on Learning Representations. 2023.

---

> ### Author Response · Authors · 2023-11-18
> **Response to Reviewer yUBQ (Part 1)**
>
> Thank you for the constructive comments.
>
> ### **W1** Whether reproducing the results of FedBuff should be considered as a contribution.
> We are sorry for the confusion. We want to emphasize that we never claim to reproduce the empirical results of FedBuff as one of our contributions. Instead, we aim to provide a new theoretical analysis of FedBuff (we obtain the convergence results with fewer assumptions and a slightly tighter bound on the asynchronous delay term). Also, see **Q1** for why providing a new theoretical analysis of FedBuff.
>
>
> ### **W2** The overall contribution of the proposed method.
> While we understand your concern, we believe that the contributions of a paper should depend on whether we solved the problem or provided new perspectives, not simply on whether the algorithm component is new or complicated. We believe our algorithmic design is intuitive, our experiments suggest its effectiveness and our analysis is non-trivial. Specifically, we want to highlight two things:
> * SAGA is a centralized variance reduction optimization method by maintains the previously computed gradients for each data point for reducing the stochastic variance. Our proposed CA$^2$FL is applying the variance reduction calibration globally at the server to reduce the global variance caused by heterogeneous data. Thus CA$^2$FL is not simply steming the idea of SAGA and applying it into asynchronous FL settings.
> * The theoretical analysis of SAGA relies on properties of unbiased incremental gradients for the variance reduction proposes, but our CA$^2$FL does not adhere to these properties due to the asynchronous gradient delay. Therefore, though our cached idea seems likely to stem from SAGA to FedBuff, the theoretical analysis is highly non-trivial and different from stemming the proof sketch of SAGA to the analysis of FedBuff.
>
> ### **W3** The assumption about the maximum gradient delay bound is used.
>
> The convergence analysis in [R1] (especially the heterogeneous distributed settings in Theorem 11 which aligns with the data heterogeneity setting in FL) actually keeps both the maximum and average gradient delay term without the additional bounded gradient assumption. In this sense, under the same set of assumptions, [R1] still requires the maximum gradient delay bound just like our analysis in the data heterogeneity setting.
>
> Furthermore, even in the homogeneous settings (global variance $\sigma_g=0$), a similar maximum delay dependency is shown in [R1] (Theorem 6 and Corollary 7) when adopting our original constant stepsize (and without the bounded gradient assumption) in the convergence rate. In Theorem 8 of [R1], the maximum delay irrelevant convergence results are obtained by choosing delay adaptive stepsizes.
>
> Please refer to the response to **Q2** for the question about the state delay.

---

> ### Author Response · Authors · 2023-11-18
> **Response to Reviewer yUBQ (Part 2)**
>
> ### **Q1** The novelty of the convergence analysis of FedBuff.
>
> As we have mentioned in responding **W1**, the contribution of presenting a new analysis of FedBuff lies in 1) fewer assumptions and 2) a slightly tighter bound on the asynchronous delay term.
>
> Besides this, we reproduce the analysis since we found that there are minor technical issues with the original convergence analysis in the FedBuff paper: in section D.1, Proof of Theorem 1, the equation (20) does not hold due to the misapplying inequalities. Following equation (20), there should be considering the probability and expectation of client participation, which results in larger local steps dependency in the convergence analysis of FedBuff.
>
> We have emailed and discussed their proof with the authors of FedBuff, and they have also admitted the issue. Since the FedBuff's authors have not updated and revised their paper yet, we did not directly cite and use their results in our main paper. Instead, we provide a detailed analysis with fewer assumptions and obtain the desired results as a backbone for our proposed method.
>
>
> ### **Q3** The convergence rate comparison with FedBuff and FedAsync.
> * Compare with FedAsync: the original analysis for FedAsync needs an assumption of weak convexity while we focus on a general non-convex setting. Therefore, we do not draw comparisons with FedAsync due to this fundamental difference in the underlying assumptions of the analysis.
> * We compare our analysis with (corrected) FedBuff (see Q1 for details). For simplicity, we let the notation align: $T$ is the total global round, $K$ is the number of local steps, and $M$ is the buffer size. Again, we want to emphasize that compared with FedBuff, we eliminate the assumption of bounded gradient and we obtain a slightly tighter bound on the asynchronous delay term.
>
>
> |Method|Rate|Bounded Gradident Assumption|
> |--|--|--|
> | FedBuff (corrected) | $\mathcal{O}(\frac{1}{\sqrt{TK}} + \frac{\sqrt{K}}{\sqrt{T}M} + \frac{K(1+ \tau_\max^2)}{TM^2})$ | need |
> | FedBuff (our) | $\mathcal{O}(\frac{1}{\sqrt{TKM}} + \frac{\sqrt{K}}{\sqrt{TM}} + \frac{K \tau_\max \tau_\text{avg}+\tau_\max}{T})$ |no need |
> | CA$^2$FL (proposed) | $\mathcal{O}(\frac{1}{\sqrt{TKM}} + \frac{\tau_\max + \zeta_\max}{T})$ | no need|

---

> ### Author Response · Authors · 2023-11-18
> **Response to Reviewer yUBQ (Part 3)**
>
> ### **Q2** The state delay is closely tied to the gradient delay.
>
> We apologize for the misunderstanding in our paper, and we have made this clear in the revision. In general, the state delay and the gradient delay reflect different aspects for each client.
> * The gradient delay describes the delay during local updates and communications caused by local computational capabilities and communication bandwidth between clients and the server. If at $t$-th global round, the server receives a new update from client $i$, then the gradient delay $\tau_t^i$ describes the round difference between the current global round $t$ and the global round at which this client $i$ **begins** local training. In other words, it takes $\tau_t^i$ global rounds for client $i$ to finish local training and update the server.
>
> *  While the state delay describes the frequency of participating in local training for each client (how many global rounds has it been since the last local training for a client). No matter whether client $j$ updates or not at $t$-th global round, the state delay $\zeta_t^j$ describes the round difference between the current global round $t$ and the global round at which this client $j$ **begins** to compute the last gradient. In other words, after $t-\zeta_t^j$-th global round, client $j$ does not participate in new local training until the $t$-th global round.
>
> Of course, the state delay is closely tied to the gradient delay. If a client needs more time to finish its local training (larger $\tau^i$), then the state delay $\zeta^i$ would also be relatively large since the round interval between two participation rounds will be longer. But still, they reflect different aspects and one cannot directly replace another in the analysis.
>
> ### **Q4** The additional memory overhead.
> Indeed, our design would require additional memory overhead. However, we want to emphasize that our proposed method requires extra memory overhead on the server instead of local clients, thus it would not cause extra memory overhead for resource-constraint local devices like GPUs. In each round of global training, cache variables are used to calibrate the global model after local training by performing only numerical operations, so cache variables can be stored on disks, which are usually quite cheap for servers.
>
> Second, although the additional memory overhead is highly related to the model, the task, and the total number of clients, the extra memory costs are reasonable for the server. For example, for experiments in training ResNet-18 on CIFAR-10, with 100 clients in total. The extra memory costs on the server would be about 4.5 GB in total. (The ResNet-18 model has approximately 44.6 MB when the parameters are stored as 32-bit floats.)
>
> Moreover, in Appendix A, we have extended our proposed CA$^2$FL to a memory-efficient version, which can effectively reduce the extra memory overhead by quantizing the 32-bit cached parameters to 4 or 8 bits. This can help reduce at least 70% of memory overhead on the server.
>
> ### **Q5** Compare with recent work in the decentralized setting.
>
> Thank you for pointing out this related work. We have added discussions about this paper in the revision. SWIFT is an interesting wait-free **decentralized** FL paradigm that also involves a caching idea by storing the neighboring local models. Besides the difference in settings, the motivation behind storing the latest models (updates) are quite different:
> * For SWIFT, each client stores a copy of their neighbors' model and checks whether there are new model updates from neighbors before model averaging. Thus, a client does not need to wait for all of their neighboring client models for communication and aggregation, and it comes to a "wait-free" updating scheme.
> * For CA$^2$FL, storing the latest model on the server is for tackling the convergence degradation caused by the joint effect of data heterogeneity, as the cached variables can help calibrate the global aggression. The characteristic of "wait-free" or "asynchronous" is obtained by the asynchronous FL framework instead of storing the cached variable.

---

> > ### Comment · Reviewer_yUBQ · 2023-11-20
> > **Rebuttal Response**
> >
> > Thank you to the authors for your wonderful rebuttal. I appreciate each response to the weaknesses I listed and I agree with the majority of them. Your reply alleviated W1 and W3, and I understand the response to W2. While I still think the algorithmic advancement is a little incremental, it is hard to argue with the improved performance.
> >
> > I have revised my evaluation accordingly.

---

> > > ### Author Response · Authors · 2023-11-21
> > > **Thanks for your review and response**
> > >
> > > We are glad to hear that our response has addressed your concerns. Thank you for raising the score and thanks for your constructive comments and suggestions on our paper.

---

### Official Review · Reviewer_3At2 · 2023-11-05

**Soundness:** 3 good
**Presentation:** 3 good
**Contribution:** 3 good
**Rating:** 8
**Confidence:** 5

**Summary:**

The manuscript tackles the problem of data heterogeneity in asynchronous federated learning. The authors show that the asynchronous delay can negatively affect the convergence of the learning process, especially when the data across clients are highly non-i.i.d. To address this issue, they propose a novel method called  Cached-Aided Asynchronous FL, which uses cached updates from each client to calibrate the global update at the server. They prove that the proposed method can improve the convergence rate under nonconvex stochastic settings and demonstrate its performance on several vision and language tasks. Moreover, the authors present a convergence analysis of FedBuff algorithm, a prior asynchronous FL algorithm, under non-i.i.d. distributed data across clients.

**Strengths:**

The paper has several strengths that make it an incremental contribution to the field of asynchronous FL. First, the paper presents a novel and unified convergence analysis of FedBuff algorithm. The paper also makes fewer assumptions (similar to Koloskova et al.) and provides slightly tighter bounds on the asynchronous delay term than previous works. Second, the paper proposes a novel method that improves the convergence rate under nonconvex settings. Third, the paper demonstrates the superior performance on both vision and language tasks.

**Weaknesses:**

The paper has some limitations that could be further addressed. First, the paper only evaluates its method on relatively simple vision and language tasks, such as CIFAR. It would be interesting to see how the method performs on more challenging data sets, such as ImageNet. Second, the paper only considers classification tasks for language, which may not fully capture the complexity and diversity of natural language processing. It would be worthwhile to explore how the method works on other language tasks, such as generation, translation, summarization, etc.

**Questions:**

The paper “Unbounded Gradients in Federated Learning with Buffered Asynchronous Aggregation” also improves the analysis of FedBuff. I recommend comparing your FedBuff analysis with this work as well.

---

> ### Author Response · Authors · 2023-11-18
> **Response to Reviewer 3At2**
>
> Thank you for the constructive comments.
>
> ### **W1** Add experiments for ImageNet
>
> Thanks for your suggestion. We believe that it is crucial to conduct scaling-up experiments to verify the proposed method. However, due to time constraints,  (and Imagenet-1K data is rarely evaluated in previous FL works), we add the experiments on fine-tuning TinyImageNet with two ResNet models.
>
> ResNet-18:
> |FedAsync |FedBuff | CA$^2$FL (propsoed)|
> |--|--|--|
> |50.74 $\pm$ 1.08 |55.55 $\pm$ 0.37 |**56.17 $\pm$ 0.23**|
>
> ResNet-34:
> |FedAsync |FedBuff | CA$^2$FL (propsoed)|
> |--|--|--|
> | 54.13 $\pm$ 1.12 |61.85 $\pm$ 0.38 |**62.51 $\pm$ 0.20**|
>
> The two tables above show that CA$^2$FL achieves higher accuracy than FedBuff and FedAsync, and this further verifies the effectiveness of our proposed method.
>
>
> ### **W2** Add experiments for NLG tasks
> Thank you for the suggestion. We agree that it would be more interesting to consider generation tasks for language models. We add a generation task for fine-tuning the GPT-2(small) model on the E2E NLG dataset. We report the validation loss after 50 global rounds of fine-tuning using LoRA. From the table below, we can observe that CA$^2$FL achieves a lower loss on the validation set than FedBuff and FedAsync, which demonstrates its ability on the generative language task.
> |FedAsync |FedBuff | CA$^2$FL (propsoed) |
> |--|--|--|
> |0.1533 $\pm$ 0.0438 | 0.1241 $\pm$ 0.0110|**0.1025  $\pm$ 0.0004**|
>
> ### **Q1**  Comparing the FedBuff analysis with one related work
> Thank you for pointing out this related work. We have added discussions and comparisons in the revision. Specifically,
> * Our original FedBuff analysis obtains a convergence rate of
> $\frac{1}{T} \sum_{t=1}^{T} \mathbb{E}\left\| \nabla f(x_t) \right\|^2 = \mathcal{O} \left( \frac{(( f_1 - f_*)+ \sigma^2)}{\sqrt{TKM}} \right) + \mathcal{O} \left( \frac{\sigma^2 + K\sigma_g^2}{TK} \right) + \mathcal{O} \left( \frac{\sqrt{K}\sigma_g^2}{\sqrt{TM}} \right) + \mathcal{O} \left( \frac{K \tau_{\text{max}} \tau_{\text{avg}} \sigma_g^2 + \tau_{\text{max}}\sigma^2}{T} \right).$
>   * If we only consider the $T$ and $\tau$ related terms, we obtain a similar convergence rate of $\mathcal{O} (\frac{1}{\sqrt{T}}) + \mathcal{O} (\frac{\tau_{\max} \tau_{\text{avg}}}{T})$ compared to $\mathcal{O} (\frac{1}{\sqrt{T}}) + \mathcal{O} (\frac{\tau_{\max}^2}{T})$ in [1].
>   * If we consider the convergence rate w.r.t. $T, K, M$ and $\tau$ related terms, when $M>K$, we obtain a similar convergence rate of $\mathcal{O} (\frac{1}{\sqrt{T}}) + \mathcal{O} (\frac{K \tau_{\max} \tau_{\text{avg}}}{T})$ compared to $\mathcal{O} (\frac{1}{\sqrt{T}}) + \mathcal{O} (\frac{K \tau_{\max}^2}{T})$ in [1].
>
> * Moreover, the convergence rate is highly related to the learning rate choosing. For example, when adopting a different learning rate as in Theorem 3.4, our convergence rate can match the result in [1] w.r.t. $T, K, M$ and $\tau$ without any further constraints: following Equation (C.15) in Appendix, choosing $\eta=\mathcal{O} (1)$ and $\eta_l = \mathcal{O} (\frac{1}{\sqrt{T}K})$, then we get
> $\frac{1}{T} \sum_{i=1}^T \mathbf{E}[\|\nabla f(x_t)\|^2] = \mathcal{O} (\frac{(f_1 - f_*)}{\sqrt{T}}) + \mathcal{O} (\frac{\sigma^2 }{\sqrt{T}KM}) + \mathcal{O} (\frac{\sigma_g^2 }{\sqrt{T}M}) + \mathcal{O} (\frac{\tau_{\max}\tau_{avg} \sigma_g^2}{TM}) + \mathcal{O} (\frac{\tau_{\max} \sigma^2}{TKM}) + \mathcal{O} (\frac{\sigma^2 + K\sigma_g^2}{TK^2}).$
> If we look at the $T$ and delay related terms, our rate would be $\mathcal{O}(\frac{1}{\sqrt{T}}) + \mathcal{O}(\frac{\tau_{\max}\tau_{avg}}{TM})$, and this is slighlty better on the non-dominant $\mathcal{O}(\frac{\tau_{\max}\tau_{avg}}{TM})$ term than the rate in [1].
>
> [1] Unbounded Gradients in Federated Learning with Buffered Asynchronous Aggregation

---

> > ### Comment · Reviewer_3At2 · 2023-11-19
> > **Revised Evaluation**
> >
> > Thanks for addressing my question. I am willing to revise my evaluation.

---

> > > ### Author Response · Authors · 2023-11-19
> > > **Thanks for your review and response**
> > >
> > > We are glad that our responses have addressed your concerns and questions. Thank you for raising the score and for your constructive suggestions.

---

### Meta-Review · Area_Chair_jJQD · 2023-12-11

**Metareview:**

The paper provides a new theoretical convergence analysis of FedBuff, a common method for realistic asynchronous federated learning in the important setting of heterogeneous data. It also shows improvements to the algorithm possible by caching, again focusing on the high heterogeneity case.

Reviewers were consistently positive about the paper, and liked the overall contributions made. This includes convincing experimental results, relevant theory, sufficient novelty and a clearly written paper, overall well-deserving acceptance.

We hope the authors will incorporate the several minor points mentioned by the reviewers during the discussions.

**Justification For Why Not Higher Score:**

One could consider lifting to the level of spotlight, though the level of novelty (and closeness to existing algorithms and analysis) might leave it more likely in the poster category.

**Justification For Why Not Lower Score:**

All reviewers were positive about the paper

---

### Decision · Program_Chairs · 2024-01-16

Accept (poster)